# Chromosome organization by one-sided and two-sided loop extrusion

Edward J Banigan[1,2†], Aafke A van den Berg[1,2†], Hugo B Brandão[3†], John F Marko[4], Leonid A Mirny[1,2]*

[1]Institute for Medical Engineering & Science, Massachusetts Institute of Technology, Cambridge, United States; [2]Department of Physics, Massachusetts Institute of Technology, Cambridge, United States; [3]Harvard Graduate Program in Biophysics, Harvard University, Cambridge, United States; [4]Departments of Molecular Biosciences and Physics & Astronomy, Northwestern University, Evanston, United States

**Abstract** SMC complexes, such as condensin or cohesin, organize chromatin throughout the cell cycle by a process known as loop extrusion. SMC complexes reel in DNA, extruding and progressively growing DNA loops. Modeling assuming two-sided loop extrusion reproduces key features of chromatin organization across different organisms. In vitro single-molecule experiments confirmed that yeast condensins extrude loops, however, they remain anchored to their loading sites and extrude loops in a 'one-sided' manner. We therefore simulate one-sided loop extrusion to investigate whether 'one-sided' complexes can compact mitotic chromosomes, organize interphase domains, and juxtapose bacterial chromosomal arms, as can be done by 'two-sided' loop extruders. While one-sided loop extrusion cannot reproduce these phenomena, variants can recapitulate in vivo observations. We predict that SMC complexes in vivo constitute effectively two-sided motors or exhibit biased loading and propose relevant experiments. Our work suggests that loop extrusion is a viable general mechanism of chromatin organization.

*For correspondence:
lmirny@gmail.com

†These authors contributed equally to this work

Competing interests: The authors declare that no competing interests exist.

## Introduction

Structural Maintenance of Chromosomes (SMC) complexes are ring-like protein complexes that are integral to chromosome organization in organisms ranging from bacteria to humans. SMC complexes linearly compact mitotic chromosomes in metazoan cells (*Gibcus et al., 2018*; *Hirano et al., 1997*; *Hirano and Mitchison, 1994*; *Ono et al., 2003*; *Shintomi et al., 2017*; *Shintomi et al., 2015*), maintain topologically associated domains (TADs) in interphase vertebrate cells (*Gassler et al., 2017*; *Haarhuis et al., 2017*; *Rao et al., 2017*; *Sanborn et al., 2015*; *Schwarzer et al., 2017*; *Wutz et al., 2017*), and juxtapose the arms of circular chromosomes in bacteria (*Marbouty et al., 2015*; *Tran et al., 2017*; *Wang et al., 2017*; *Wang et al., 2015*). In each of these processes, SMC complexes form chromatin loops. These diverse chromosome phenomena are hypothesized to be driven by a common underlying physical mechanism by which SMC complexes processively extrude chromatin or DNA loops (*Alipour and Marko, 2012*; *Bürmann and Gruber, 2015*; *Fudenberg et al., 2017*; *Fudenberg et al., 2016*; *Goloborodko et al., 2016a*; *Goloborodko et al., 2016b*; *Gruber, 2014*; *Nasmyth, 2001*; *Riggs, 1990*; *Sanborn et al., 2015*; *Wang et al., 2017*; *Wang et al., 2015*). However, it is not known what molecular-level requirements loop extrusion must satisfy in order to robustly reproduce the 3D chromosome structures observed in these in vivo phenomena.

The loop extrusion model posits that a loop-extruding factor (LEF), such as condensin, cohesin, or a bacterial SMC complex (bSMC) is in part comprised of two connected motor subunits that bind to chromatin and form a small chromatin loop by bridging two proximal chromatin segments. The SMC

**eLife digest** The different molecules of DNA in a cell are called chromosomes, and they change shape dramatically when cells divide. Ordinarily, chromosomes are packaged by proteins called histones to make thick fibres called chromatin. Chromatin fibres are further folded into a sparse collection of loops. These loops are important not only to make genetic material fit inside a cell, but also to make distant regions of the chromosomes interact with each other, which is important to regulate gene activities. The fibres compact to prepare for cell division: they fold into a much denser series of loops. This is a remarkable physical feat in which tiny protein machines wrangle lengthy strands of DNA.

A process called loop extrusion could explain how chromatin folding works. In this process, ring-like protein complexes known as SMC complexes would act as motors that can form loops. SMC complexes could bind a chromatin fibre and reel it in to form the loops, with the density of loops increasing before cell division to further compact the chromosomes. Looping by SMC complexes has been observed in a variety of cell types, including mammalian and bacterial cells. From these studies, loop extrusion is generally assumed to be 'two-sided'. This means that each SMC complex reels in the chromatin on both sides of it, thus growing the chromatin loop.

However, imaging individual SMC complexes bound to single molecules of DNA showed that extrusion can be asymmetric, or 'one-sided'. These observations show the SMC complex remains anchored in place and the chromatin is reeled in and extruded by only one side of the complex. So Banigan, van den Berg, Brandão et al. created a computer model to test whether the mechanism of one-sided extrusion could produce chromosomes that are organised, compact, and ready for cell division, like two-sided extrusion can.

To answer this question, Banigan, van den Berg, Brandão et al. analysed imaging experiments and data that had been collected using a technique that captures how chromatin fibres are arranged inside cells. This was paired with computer simulations of chromosomes bound by SMC protein complexes. The simulations and analysis found that the simplest one-sided loop extrusion complexes generally cannot reproduce the same patterns of chromatin loops as two-sided complexes. However, a few specific variations of one-sided extrusion can actually recapitulate correct chromatin folding and organisation.

These results show that some aspects of chromosome organization can be attained by one-sided extrusion, but many require two-sided extrusion. Banigan, van den Berg, Brandão et al. explain how the simulated mechanisms of loop extrusion could be consistent with seemingly contradictory observations from different sets of experiments. Altogether, they demonstrate that loop extrusion is a viable general mechanism to explain chromatin organisation, and that it likely possesses physical capabilities that have yet to be observed experimentally.

complex progressively enlarges the loop by reeling chromatin from outside the loop into the growing loop (*Alipour and Marko, 2012*; *Nasmyth, 2001*; *Riggs, 1990*). To reel in chromatin from both sides of the complex, each motor subunit of the LEF translocates in opposite directions, away from the initial binding site (*Alipour and Marko, 2012*; *Fudenberg et al., 2016*; *Goloborodko et al., 2016a*; *Goloborodko et al., 2016b*; *Sanborn et al., 2015*). This 'two-sided' extrusion model recapitulates experimental observations of mitotic chromosome compaction and resolution, interphase TAD and loop formation, and juxtaposition of bacterial chromosome arms (*Alipour and Marko, 2012*; *Fudenberg et al., 2016*; *Goloborodko et al., 2016a*; *Goloborodko et al., 2016b*; *Miermans and Broedersz, 2018*; *Sanborn et al., 2015*; *Wang et al., 2017*; *Wang et al., 2015*). However, until recently, loop extrusion by SMC complexes had not been directly observed.

Recent in vitro single-molecule experiments have imaged loop extrusion of DNA by individual SMC condensin and cohesin complexes, demonstrating that yeast, human, and *Xenopus* condensin and *Xenopus* cohesin complexes extrude DNA loops in an ATP-dependent, directed manner at speeds on the order of 1 kb/s (*Davidson et al., 2019*; *Ganji et al., 2018*; *Golfier et al., 2020*; *Kim et al., 2019*; *Kong et al., 2020*). Strikingly, however, yeast condensins (*Ganji et al., 2018*) and a significant fraction of both human and *Xenopus* SMC complexes (*Golfier et al., 2020*; *Kong et al., 2020*) reel in DNA from only one side, while the other side remains anchored to its DNA loading

site. This contrasts with prior observations in bacteria demonstrating the direct involvement of SMC complexes in two-sided loop extrusion in vivo (*Tran et al., 2017*; *Wang et al., 2017*). One-sided extrusion also conflicts with existing versions of the loop extrusion model, which generally assume that extrusion is two-sided (*Alipour and Marko, 2012*; *Fudenberg et al., 2016*; *Goloborodko et al., 2016a*; *Goloborodko et al., 2016b*; *Miermans and Broedersz, 2018*; *Sanborn et al., 2015*). Furthermore, recent theoretical work shows that purely 'one-sided' loop extrusion, as it has been observed in vitro so far, is intrinsically far less effective in linearly compacting DNA than two-sided extrusion (*Banigan and Mirny, 2019*). Thus, we investigated the extent to which one-sided loop extrusion might impact the 3D structure of chromosomes and whether variants of one-sided loop extrusion can recapitulate in vivo observations. In particular, we focus on three chromosome organization phenomena that are driven by SMC complexes: (1) mitotic chromosome compaction and resolution, (2) interphase chromosome domain formation, and (3) juxtaposition of bacterial chromosome arms. These three phenomena encompass the major physical processes associated with chromosome organization by SMC complexes: compaction and segregation, *cis* loop formation and linear scanning, and progressive juxtaposition of DNA flanking a loading site.

## Mitotic chromosome compaction and resolution

The SMC condensin complex in metazoan cells plays a central role in mitotic chromosome compaction and segregation (*Charbin et al., 2014*; *Hagstrom et al., 2002*; *Hirano, 2016*; *Hirano et al., 1997*; *Hirano and Mitchison, 1994*; *Hudson et al., 2003*; *Nagasaka et al., 2016*; *Ono et al., 2003*; *Piskadlo et al., 2017*; *Saka et al., 1994*; *Shintomi et al., 2017*; *Shintomi et al., 2015*; *Steffensen et al., 2001*; *Strunnikov et al., 1995*). In mitotic chromosomes, electron microscopy reveals that chromatin is arranged in arrays of loops (*Earnshaw and Laemmli, 1983*; *Maeshima et al., 2005*; *Marsden and Laemmli, 1979*; *Paulson and Laemmli, 1977*). This results in dramatic linear compaction of the chromatin fiber into a polymer brush with a > 100 fold shorter backbone (*Guacci et al., 1994*; *Lawrence et al., 1988*; *Trask et al., 1989*). Fluorescence imaging and Hi-C show that these loops maintain the linear ordering of the genome (*Gibcus et al., 2018*; *Naumova et al., 2013*; *Strukov and Belmont, 2009*; *Trask et al., 1993*). Together, these features may facilitate the packaging, resolution, and segregation of chromosomes during mitosis by effectively shortening and disentangling chromatids (*Brahmachari and Marko, 2019*; *Eykelenboom et al., 2019*; *Goloborodko et al., 2016a*; *Green et al., 2012*; *Marko, 2009*; *Nagasaka et al., 2016*; *Sakai et al., 2018*; *Sakai et al., 2016*). Each of these experimental observations is reproduced by the two-sided loop extrusion model, in which dynamic loop-extruding condensins collectively form arrays of reinforced loops by locally extruding chromatin until encountering another condensin (*Goloborodko et al., 2016a*; *Goloborodko et al., 2016b*). The simplest one-sided loop extrusion process, in contrast, can only linearly compact chromosomes 10-fold because it leaves unlooped (and thus, uncompacted) polymer gaps between loop extruders (*Banigan and Mirny, 2019*); it is unclear whether 10-fold compaction is sufficient for robust chromosome segregation. Nonetheless, variants of one-sided loop extrusion in which loop extruders are effectively two-sided may robustly compact mitotic chromosomes (*Banigan and Mirny, 2019*). This raises the question of what abilities an individual one-sided loop extruder must possess to compact and spatially resolve chromosomes.

## Interphase domain formation

In interphase in vertebrate cells, Hi-C reveals that the SMC cohesin complex is responsible for frequent but transient loop formation, which results in regions of high intra-chromatin contact frequency referred to as TADs (*Dixon et al., 2012*; *Gassler et al., 2017*; *Haarhuis et al., 2017*; *Nora et al., 2012*; *Rao et al., 2017*; *Rao et al., 2014*; *Schwarzer et al., 2017*; *Sexton et al., 2012*; *Sofueva et al., 2013*). These regions are bordered by convergently oriented CTCF protein binding sites (*de Wit et al., 2015*; *Guo et al., 2015*; *Rao et al., 2014*; *Sanborn et al., 2015*; *Vietri Rudan et al., 2015*), which act as obstacles to loop extrusion and translocation of cohesin (*Busslinger et al., 2017*; *de Wit et al., 2015*; *Fudenberg et al., 2016*; *Nora et al., 2017*; *Sanborn et al., 2015*; *Wutz et al., 2017*). The two-sided loop extrusion model explains the emergence of TADs and their 'corner peaks' (or 'dots') and 'stripes' (sometimes called 'lines', 'tracks' or 'flames') in Hi-C maps as an average collective effect of multiple cohesins dynamically extruding

chromatin loops and stopping at the CTCF boundaries (*Fudenberg et al., 2016*; *Sanborn et al., 2015*; reviewed in *Fudenberg et al., 2017*). Existing models for loop extrusion during interphase have assumed LEFs with two mobile subunits, whether they be active or inactive (*Alipour and Marko, 2012*; *Benedetti et al., 2017*; *Brackley et al., 2017*; *Fudenberg et al., 2016*; *Sanborn et al., 2015*; *Yamamoto and Schiessel, 2017*). While it is clear that a one-sided LEF will necessarily leave an unlooped gap between its initial loading site and one of the CTCF boundary elements, the extent to which one-sided loop extrusion can recapitulate the experimental observations remains entirely unexplored.

## Bacterial chromosome arm juxtaposition

In bacteria, SMC complexes and homologs play an important role in the maintenance of proper chromosome organization and efficient chromosomal segregation (*Britton et al., 1998*; *Jensen and Shapiro, 1999*; *Moriya et al., 1998*; *Sullivan et al., 2009* and others). In *Bacillus subtilis* and *Caulobacter crescentus*, the circular chromosome exhibits enhanced contact frequency between its two chromosomal arms (often called 'replichores'), as shown by Hi-C (*Le et al., 2013*; *Marbouty et al., 2015*). This signal is dependent on the bacterial SMC complex (bSMC) (*Marbouty et al., 2015*; *Wang et al., 2015*). Experiments show that bSMC is loaded at a bacterial *parS* site near the origin of replication, and then, while bridging the two arms, actively and processively moves along the chromosome, thus juxtaposing or 'zipping' the arms together (*Minnen et al., 2016*; *Tran et al., 2017*; *Wang et al., 2018*; *Wang et al., 2017*). The symmetry of the juxtaposed chromosome arms implies that bSMC should be a two-sided LEF (*Brandão et al., 2019*; *Wang et al., 2017*). Indeed, previous modeling has shown that pure one-sided loop extrusion produces contact maps that differ from experimental observations (*Miermans and Broedersz, 2018*). However, it is unknown whether variations of one-sided extrusion can properly juxtapose the arms of a circular bacterial chromosome.

## Objectives

Two-sided loop extrusion models (*Brandão et al., 2019*; *Fudenberg et al., 2017*; *Fudenberg et al., 2016*; *Goloborodko et al., 2016a*; *Goloborodko et al., 2016b*; *Sanborn et al., 2015*) can account for the various chromosome organization phenomena described above, but in vitro single-molecule experiments suggest that at least some SMC complexes are one-sided LEFs. We therefore investigate whether a mechanism of one-sided loop extrusion can account for in vivo observations of 3D chromatin organization, as listed above, namely metazoan mitotic chromosome compaction and resolution, interphase chromatin organization in vertebrate cells, and juxtaposition of bacterial chromosome arms. To study these processes, we construct a model for one-sided loop extrusion and simulate the collective dynamics of SMC complexes and chromatin in these three distinct scenarios. We also explore several one-sided extrusion variants. By comparing our results to experimental data, we find that pure one-sided loop extrusion fails to capture most of the in vivo phenomenology. However, simple variants of the one-sided model that make loop extrusion effectively two-sided or otherwise suppress the formation of unlooped chromatin gaps can restore the emergent features of chromatin organization observed in experiments.

## Model
### Model for loop extrusion

In our model, loop extrusion is performed by loop-extruding factors (LEFs), which may be a single SMC complex, a dimer of SMC complexes, or any other oligomer of SMC complexes. A LEF is comprised of two subunits, which can either be active or inactive. Each active subunit can processively translocate along the chromatin fiber, thus creating and enlarging the chromatin (or DNA) loop between the subunits (*Figure 1a*). An inactive subunit can either be anchored or passively slide/diffuse along the fiber, depending on the specific model (see below).

In existing simulation models of loop extrusion (*Alipour and Marko, 2012*; *Brandão et al., 2019*; *Fudenberg et al., 2016*; *Goloborodko et al., 2016a*; *Goloborodko et al., 2016b*; *Miermans and Broedersz, 2018*; *Sanborn et al., 2015*), LEFs are 'two-sided,' that is they have two active subunits that on average grow a chromatin loop by translocating in opposing directions (*Figure 1b*). Here, we consider 'one-sided' LEFs that have one active subunit and one inactive (passive) subunit.

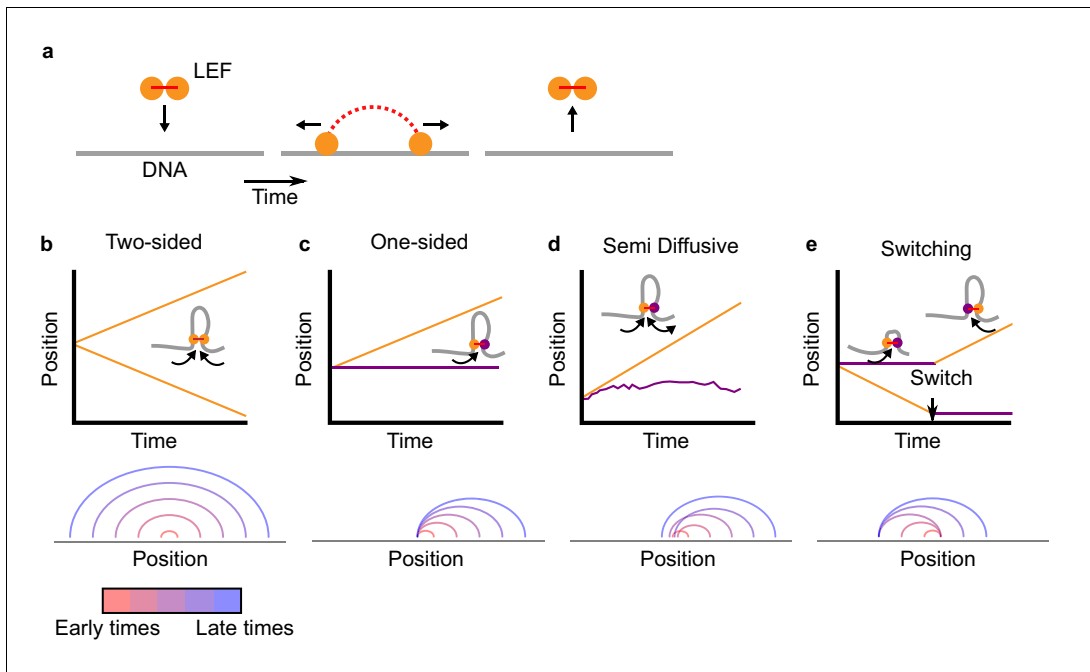

**Figure 1.** Two-sided loop extrusion and variants of one-sided loop extrusion. (**a**) A schematic of the loop extrusion model. The two subunits of the LEF bind to sites on a one-dimensional lattice representing DNA/chromatin. Over time, the subunits may translocate along DNA, and the LEF eventually unbinds from DNA. In 3D polymer simulations, the two subunits remain in spatial proximity (in 3D) while translocating along DNA (in 1D), thereby extruding loops. (**b**) *Top:* The positions of the two LEF subunits versus time for a two-sided LEF. *Inset:* Cartoon of a two-sided LEF on DNA extruding a loop. *Bottom*: Arch diagram showing the positions of the LEF subunits from early times (red) to late times (blue). (**c**) *Top:* Time trace of a one-sided LEF with inset schematic. In the example in the schematic, the active subunit is on the left, but in the model LEFs are loaded with random orientations. *Bottom:* Arch diagram for a one-sided LEF, where the left subunit is stationary (passive). (**d**) *Top:* The positions of the two LEF subunits versus time for the semi-diffusive model. The speed of loop growth increases as the loop grows because the entropic cost of loop growth most strongly affects small loops. *Bottom:* Arch diagram for the semi-diffusive model, where the left subunit is diffusive. (**e**) *Top:* Schematic and a time trace of the switching model. *Bottom:* Example of an arch diagram for a LEF in the switching model. Note: the arch diagrams do not correspond to the time traces, but rather, they are illustrative examples.

LEFs in our one-sided extrusion model have binding and translocation dynamics that mimic turnover and translocation of SMC complexes, as has been observed in experiments (*Ganji et al., 2018*; *Gerlich et al., 2006a*; *Gerlich et al., 2006b*; *Hansen et al., 2017*; *Kleine Borgmann et al., 2013*; *Kueng et al., 2006*; *Stigler et al., 2016*; *Tedeschi et al., 2013*; *Terakawa et al., 2017*; *Tran et al., 2017*; *Walther et al., 2018*; *Wang et al., 2017*; *Wutz et al., 2017*). In our model, LEFs bind to chromatin with association rate $k_{bind}$ and unbind from chromatin with dissociation rate $k_{unbind}$ (mean residence time $\tau = 1/k_{unbind}$). A LEF's active subunit translocates at speed $v$ along the chromosome, away from its passive subunit, thus growing the chromatin loop. Furthermore, LEF subunits cannot translocate through other LEF subunits unless otherwise stated; extrusion by an active LEF subunit halts when it encounters another LEF subunit. Extrusion may continue if the obstacle is removed (for example, by unbinding). This constraint is relaxed for one model variant, as described in the Results section.

The pure one-sided and two-sided loop-extrusion models are primarily controlled by two length scales, $\lambda$ and $d$ (*Banigan and Mirny, 2019*; *Fudenberg et al., 2016*; *Goloborodko et al., 2016b*). The LEF processivity $\lambda$ is given by $\lambda = qv/k_{unbind}$, where $q = 1$ or $q = 2$ for one- and two-sided, respectively; thus, one-sided LEFs with extrusion velocity $v$ grow loops at half the speed of two-sided LEFs with the same $v$ (see arch diagrams in *Figure 1b and c*, bottom). $d = L/N_b$, is the mean distance between the $N_b$ LEFs bound to the fiber of length $L$ (where $N_b = N\, k_{bind}/(k_{bind}+k_{unbind})$). For $\lambda < d$,

LEFs are sparse and on average do not meet. For $\lambda > d$, LEFs are densely loaded on the chromatin, and a translocating LEF typically encounters other LEFs.

While there are many possible variants of the one-sided loop extrusion model, we mainly focus on three general variants of one-sided loop extrusion that differ by LEF subunit translocation dynamics.

### Pure one-sided extrusion

In pure one-sided loop extrusion, the passive subunit of the bound LEF remains stationary on the chromatin fiber for the entire residence time of the LEF, while the active subunit translocates at speed $v$ away from the passive subunit. LEFs bind with a random orientation. Individual LEFs asymmetrically extrude loops, as observed in *Ganji et al. (2018)*. *Figure 1c* shows a typical trajectory and corresponding arch diagram for LEF subunits in the pure one-sided extrusion model.

### Semi-diffusive model

We also considered a model in which the active LEF subunit translocates at speed $v$, while the inactive LEF subunit stochastically diffuses (slides) along the fiber. This model is primarily motivated by the experimental observation of the yeast condensin 'safety belt' (*Kschonsak et al., 2017*). This condensin component is thought to anchor the LEF in place as it extrudes loops in a one-sided manner, but the safety belt can be released via protein alterations, allowing the passive subunit of the SMC complex to diffuse along DNA (*Ganji et al., 2018*; *Kschonsak et al., 2017*). In addition, we note that several in vitro experiments have imaged cohesins and condensins diffusively translocating along naked DNA with diffusion coefficients of $D$ = 0.001–4 $\mu$m$^2$/s (or $D$ = 0.01–35 kb$^2$/s) (*Davidson et al., 2016*; *Kanke et al., 2016*; *Kim and Loparo, 2016*; *Kim et al., 2019*; *Stigler et al., 2016*; *Terakawa et al., 2017*).

In the model, the inactive subunit stochastically translocates by taking diffusive steps in either direction. The stepping rate in each direction is modulated by the entropic penalty for polymer loop formation (see Materials and methods). As a result of this effect, the sliding tends to shrink small loops, while having little effect on large loops. A typical trajectory and arch diagram for the subunits of a semi-diffusive LEF are shown in *Figure 1d*.

To evaluate the importance of passive extrusion as compared to active extrusion, we study loop extrusion as a function of the scaled diffusive stepping rate. This quantity is the ratio, $v_{\text{diff}}/v$, of the characteristic diffusive stepping rate, $v_{\text{diff}}$, to the active loop extrusion speed, $v$. $v_{\text{diff}}/v < 1$ indicates that diffusive stepping is slow as compared to active stepping, while $v_{\text{diff}}/v > 1$ indicates that diffusive stepping is relatively rapid. The scaled diffusive stepping rate may be converted to a diffusion coefficient by $D = a\,v$, where $a$ is the length of a lattice site.

### Switching model

As another alternative model, we consider a scenario in which LEFs are instantaneously one-sided (i. e., one subunit is active and the other is inactive and stationary), but stochastically switch which subunit actively translocates. This model captures the dynamics of a proposed mechanism dubbed 'asymmetric strand switching' (see Figure 2d in *Hassler et al., 2018*). As described in *Marko et al. (2019)*, switching could be achieved through a stochastic segment/loop-capture mechanism. In our model, switches occur at rate $k_{\text{switch}}$; by switching, inactive subunits become active and vice versa. Thus, LEF subunits have trajectories similar to the one shown in *Figure 1e*, top panel, and loops grow as shown in the arch diagram at the bottom of *Figure 1e*. Although not yet observed experimentally, we hypothesize that switching activity of SMC complexes could potentially be induced by exchange of subunits within the SMC complex, different solution conditions, or post-translational or genetic modifications, all of which can alter SMC complex behavior in experiments (*Eeftens et al., 2017*; *Elbatsh et al., 2019*; *Ganji et al., 2018*; *Keenholtz et al., 2017*; *Kleine Borgmann et al., 2013*; *Kschonsak et al., 2017*).

We explore the switching model by varying the switching rate scaled by either the dissociation rate $k_{\text{unbind}}$ (for the eukaryotic chromosome models) or the chromosome traversal rate $v/L$ (for the bacterial chromosome model). For the eukaryotic models, the dimensionless ratio $k_{\text{switch}}/k_{\text{unbind}}$ determines the mean number of switches before a LEF unbinds from the chromatin fiber (*Banigan and Mirny, 2019*). For $k_{\text{switch}}/k_{\text{unbind}} < 1$, switches rarely occur and LEF trajectories typically appear to be pure one-sided. In contrast, for $k_{\text{switch}}/k_{\text{unbind}} > 1$, the active and inactive LEF subunits

may frequently switch before unbinding chromatin, and trajectories appear as in *Figure 1e*, top panel. For bacteria, the dimensionless quantity $k_{switch}L/v$ is a dimensionless measure of the switching rate, chosen because chromosome-traversing bacterial SMC complexes (like *B. subtilis* SMC complexes) do not have a well defined unbinding rate. When this ratio is large, switching occurs many times during chromosome traversal; when it is small, switching is rare.

## Models for 3D chromosome conformations

We investigated the degree to which the above models reproduce physiological chromosome structures via 3D polymer simulations. To do this, we coupled each of the 1D loop-extrusion models in *Figure 1* to a 3D model of a polymer chain (*Fudenberg et al., 2016*; *Goloborodko et al., 2016a*) and performed molecular dynamics simulations using OpenMM (see Materials and methods for details) (*Eastman et al., 2017*; *Eastman et al., 2013*; *Eastman and Pande, 2010*). In this coupled model, LEFs act as a bond between the two sites (monomers) to which the LEF subunits are bound; these bonds have the dynamics described for LEFs above. We simulated each of the three models, as well as several other variants, for various values of $\lambda$, $d$, $v_{diff}/v$, and either $k_{switch}/k_{unbind}$ or $k_{switch}L/v$. From these simulations, we obtain 3D polymer structures, images of compacted chromosomes and/or contact frequency (Hi-C-like) maps. By analyzing these data, we compare the models to experiments.

In addition to 3D polymer simulations, we generated contact maps semi-analytically from the 1D models of the underlying SMC dynamics. This method allowed us to explore a broad range of parameter values and assess the resulting Hi-C-like maps in a computationally inexpensive manner. The semi-analytical method is compared to the 3D polymer simulation method in Appendix 3. The semi-analytical method is not used for modeling the eukaryotic systems because the Gaussian approximation used is not appropriate for highly compacted mitotic and 'vermicelli' (*i.e.*, Wapl depletion [*Tedeschi et al., 2013*]) interphase chromosomes, which have linearly dense arrays of chromatin loops. However, as shown in Appendix 3, this method can be used to study bacterial chromosome conformations.

We analyze these models for three chromosome phenomena that depend on SMC complexes. Each of the following results sections briefly describes the scenario, explains the relevant model observables, and subsequently, explores each model variant.

# Results

## Compaction and resolution of mitotic chromosomes

### Model and observables

We determined whether variants of the one-sided loop extrusion model can explain mitotic chromosome compaction and the spatial resolution of connected sister chromatids. Experimentally, it has been shown that these phenomena are driven by the condensin complex (*Eykelenboom et al., 2019*; *Hagstrom et al., 2002*; *Hirano, 2016*; *Hirano et al., 1997*; *Hirano and Mitchison, 1994*; *Hudson et al., 2003*; *Nagasaka et al., 2016*; *Ono et al., 2003*; *Piskadlo et al., 2017*; *Shintomi et al., 2017*; *Shintomi et al., 2015*; *Steffensen et al., 2001*). During mitosis, mammalian chromosomes are linearly compacted ~1000 fold, leading to the formation of rod-like chromatids. Such compaction is thought to facilitate the spatial resolution of sister chromatids, which are connected at their centromeres.

Previous work suggests that the two-sided loop extrusion model can rapidly achieve 1000-fold linear compaction in the regime in which LEFs are densely loaded on the chromosome ($\lambda/d \geq 10$), which is expected for mitotic chromosomes in metazoan cells (*Goloborodko et al., 2016b*). With a loop extrusion speed of $v \approx 1$ kb/s (*Ganji et al., 2018*), two-sided extrusion can achieve full linear compaction within one residence time ($1/k_{unbind} \sim 2$–10 min [*Gerlich et al., 2006a*; *Terakawa et al., 2017*; *Walther et al., 2018*]) and full 3D compaction and loop maturation occurs over a few (<10) residence times (*Goloborodko et al., 2016a*), consistent with the duration of prophase and prometaphase and in vivo observations of mitotic chromosome compaction (*Eykelenboom et al., 2019*; *Gibcus et al., 2018*) and resolution (*Eykelenboom et al., 2019*).

In contrast, theoretical work has demonstrated that pure one-sided loop extrusion cannot linearly compact a chromatin fiber by more than ~10 fold (*Banigan and Mirny, 2019*). Linear compaction in

these models depends only on the dimensionless ratio of length scales $\lambda/d$ (**Banigan and Mirny, 2019**; **Goloborodko et al., 2016b**). However, the 3D structures of such chromosomes have not yet been studied, and compaction by the semi-diffusive model, switching model, and other model variants has not been comprehensively investigated. Furthermore, sister chromatid resolution by variations of the one-sided loop extrusion model has not been investigated.

We therefore performed simulations to measure linear compaction and characteristics of 3D chromosome organization of individual, compacted chromosomes. To measure linear compaction, we define the compacted fraction, $f$, as the fraction of chromosome length that is contained within looped regions and the resulting linear fold compaction as $FC = 1/(1-f)$. We measure the resulting 3D compaction by computing chromosome volume, $V$, which is expected to decrease by >2 fold during mitotic compaction (**Daban, 2003**; **Hihara et al., 2012**; **Liang et al., 2015**; **Nagasaka et al., 2016**; **Sumner, 1991**). We thus look for scenarios in which chromosomes are linearly compacted ~1000 fold and form the spatially compact rod-like arrays of chromatin loops observed in experiments (**Earnshaw and Laemmli, 1983**; **Gibcus et al., 2018**; **Guacci et al., 1994**; **Lawrence et al., 1988**; **Maeshima et al., 2005**; **Marsden and Laemmli, 1979**; **Ono et al., 2003**; **Paulson and Laemmli, 1977**; **Trask et al., 1989**; **Walther et al., 2018**).

We also characterize the ability of one-sided loop extrusion models to resolve sister chromatids connected at their centromeres. We quantify chromatid resolution by measuring the median inter-chromatid backbone distance, $\Delta R$, scaled by the polymer backbone length, $R_b$. As a supplementary metric, we also compute the inter-chromatid overlap volume, $V_o$, compared to the overlap volume without loop extrusion, $V_o^{(0)} = 3.6 \ \mu m^3$. Larger distances, $\Delta R/R_b > 1$, indicates that typical inter-chromatid distances are sufficient to prevent contacts between backbones. Larger median distance and smaller overlap are expected to contribute to the disentanglement of chromatids (**Piskadlo et al., 2017**; **Sen et al., 2016**), which facilitates chromosome segregation by preventing anaphase bridge formation (**Charbin et al., 2014**; **Green et al., 2012**; **Hagstrom et al., 2002**; **Nagasaka et al., 2016**; **Piskadlo et al., 2017**; **Steffensen et al., 2001**). Models are thus evaluated on the basis of whether compacted chromatids are fully spatially resolved.

## Pure one-sided extrusion can neither compact nor resolve chromatids

Mean-field theory predicts that pure one-sided loop extrusion can achieve at most $\approx 10$-fold linear compaction, 100-fold less than expected for mammalian mitotic chromosomes. **Figure 2c (i)** shows linear fold compaction, $FC$, as a function of $\lambda/d$ in the simulations, and results for $\lambda/d >> 1$ are consistent with the theoretical predictions (**Banigan and Mirny, 2019**). The compaction limit is due to the unavoidable presence of 'gaps' of uncompacted (unlooped) chromatin between some adjacent loops (**Figure 2c (ii)**); of the four possible orientations of adjacent translocating LEFs, $\rightarrow\rightarrow$, $\leftarrow\leftarrow$, $\rightarrow\leftarrow$, and $\leftarrow\rightarrow$, the last one necessarily leaves an unlooped gap (**Banigan and Mirny, 2019**); the mechanistic connection between gaps and deficient compaction is illustrated by simulations broadly spanning $\lambda/d$ (**Figure 2c (ii)**).

We find that the presence of unlooped gaps along the chromatin fiber additionally has severe consequences for the 3D conformations of simulated mitotic chromosomes. As shown in **Figure 2b** (left), chromosomes compacted by one-sided LEFs are more spherical, and compacted regions are interspersed with uncompacted (unlooped) chromatin fibers. Moreover, compaction by one-sided LEFs only reduces the volume, $V$, by up to 2-fold from the uncompacted volume of $V^{(0)} = 3.6 \ \mu m^3$ (**Figure 2c (iii)**). This contrasts with the structures observed and >2.5 fold 3D compaction in the two-sided loop extrusion model (**Figure 2a**, left). Moreover, adding a small number of two-sided LEFs does not close a sufficient number of gaps to achieve 1000-fold linear compaction (**Figure 2—figure supplement 1a**; **Banigan and Mirny, 2019**) or 2.5-fold volumetric compaction (**Figure 2—figure supplement 1c**) because even a small number of gaps prevents full compaction (**Figure 2—figure supplement 1b**). A fraction of >80% of two-sided LEFs is necessary for sufficient compaction and resolution. One-sided extrusion thus leads to loosely compacted chromosomes that are qualitatively different from mitotic chromosomes observed in both the two-sided loop extrusion model and in vivo.

We therefore investigated whether the inability of one-sided LEFs to compact chromosomes also impacted their ability to resolve sister chromatids. We find that one-sided LEFs can spatially resolve chromosomes that are physically linked at their centromeres, but far less effectively than two-sided

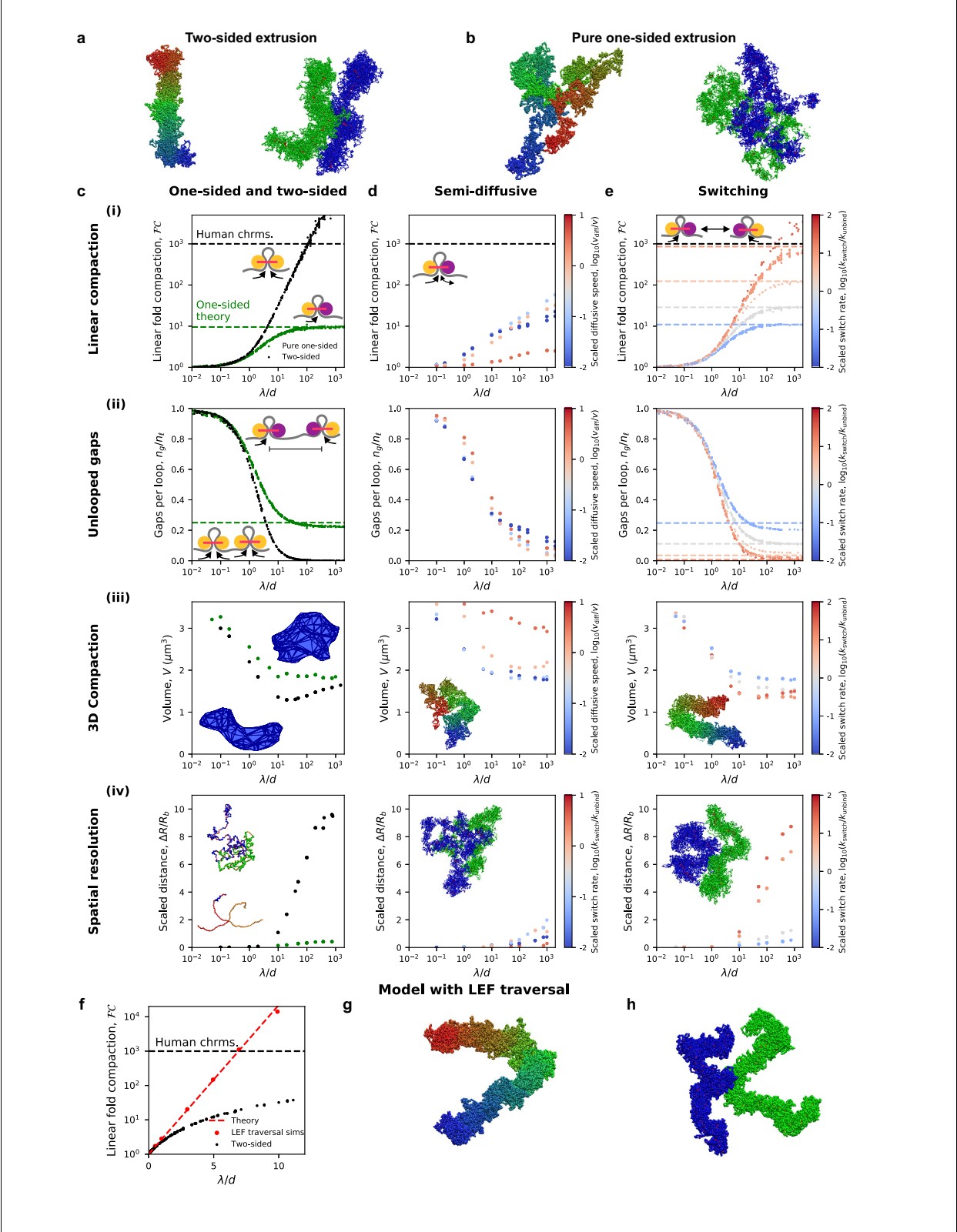

**Figure 2.** Chromosome compaction and structure in the one-sided loop extrusion model and model variants. (**a**) Simulation snapshots of chromosomes compacted (left) and spatially resolved (right) by two-sided extrusion. (**b**) Simulation snapshots showing deficient compaction (left) and resolution (right) of chromosomes with pure one-sided loop extrusion. (**c**) One-sided loop extrusion model, as compared to the two-sided model. (**i**) Linear fold compaction, *FC*, as a function of the dimensionless ratio, *λ/d*, of the processivity to the mean distance between LEFs. Pure one-sided extrusion (green)

*Figure 2 continued on next page*

*Figure 2 continued*

saturates at ≈10-fold compaction for large $\lambda/d$, as predicted by mean-field theory (green dashed line). *FC* by two-sided extrusion (black) surpasses the 1000-fold linear compaction expected for human chromosomes (black dashed line) for $\lambda/d > 50$. Insets: cartoons of extrusion of chromatin (gray) by active LEF subunits (yellow). Stationary passive subunit for one-sided LEF is purple. (ii) Number of gaps per parent loop, $n_g/n_\ell$, saturates at ≈0.25 (dashed line) as $\lambda/d$ increases in the pure one-sided model (green), as expected from theory. For two-sided extrusion, $n_g/n_\ell$ approaches 0 (black). Insets: mechanisms of gap formation and closure. (iii) Chromosome volume, *V*, decreases as $\lambda/d$ increases. *V* achieves smaller values in the two-sided model (black) than in the one-sided model (green). Insets: Images of concave hulls of simulated chromosomes compacted by one- and two-sided extrusion (top and bottom, respectively). (iv) Scaled distance, $\Delta R/R_b$, between sister chromatid backbones in one- or two-sided models. Insets: chromatid backbones in simulations of one- and two-sided extrusion (top and bottom, respectively). (d) Semi-diffusive model. (i) *FC* <1000 for $\lambda/d < 1000$. Color from blue to red indicates increasing scaled diffusive stepping speed, $v_{diff}/v$. Inset: a semi-diffusive LEF. (ii) Number of gaps per loop, $n_g/n_\ell$, versus $\lambda/d$. (iii) Compacted chromosome volume, *V*, versus $\lambda/d$. Inset: chromosome compacted by semi-diffusive LEFs with $v_{diff}/v = 1$. (iv) Scaled distance, $\Delta R/R_b$, between chromatid backbones. Inset: image of spatial resolution with $v_{diff}/v = 1$. (e) Switching model. (i) *FC* can surpass 1000-fold linear compaction for rapid scaled switching rates, $k_{switch}/k_{unbind} > 10$ (red). Simulations with large $\lambda/d$ match mean-field theoretical predictions (colored dashed lines). Inset: illustration of the model. (ii) Number of gaps per loop, $n_g/n_\ell$, with mean-field theoretical predictions (dashed lines). (iii) Compacted chromosome volume, *V*. Inset: image of compacted chromosome with $k_{switch}/k_{unbind} = 30$. (iv) Scaled distance, $\Delta R/R_b$, between chromatid backbones. Inset: spatial resolution in simulations with $k_{switch}/k_{unbind} = 30$. (f) Linear fold-compaction for a chromosome with LEFs that are able to traverse each other. Dashed line shows theoretical fold compaction, as quantified by loop coverage, $FC = e^{\lambda/d}$. (g) Simulation snapshot of chromosome compacted by LEFs that may traverse each other. (h) Simulation snapshot of chromatids resolved by LEFs that may traverse each other. Each data point is a mean quantity (see Materials and methods). Standard deviation of the mean for each point is <15% of the mean, or else smaller than the size of a data point. The online version of this article includes the following figure supplement(s) for figure 2:

**Figure supplement 1.** Compaction in model with a mix of one- and two-sided LEFs.
**Figure supplement 2.** Measures of compaction and segregation with different densities of LEFs.
**Figure supplement 3.** Loop sizes and LEF nesting explain the ineffectiveness of the semi-diffusive model.
**Figure supplement 4.** Models in which the active subunits of nested LEFs can push passive LEF subunits.
**Figure supplement 5.** Defective compaction and segregation with 3D attractive interactions.
**Figure supplement 6.** Fold linear compaction in pure one-sided extrusion models in which LEF residence times are altered by contact with other LEFs.
**Figure supplement 7.** Compaction and resolution of chromosomes with limited loop coverage.

LEFs. With one-sided extrusion, there is a small relative separation between chromatid backbones ($\Delta R/R_b$ <1, *Figure 2c* (iv)) and large overlap of chromatids ($V_o/V_o^{(0)} \approx 0.3$; *Figure 2—figure supplement 2c*). In contrast, with two-sided extrusion, there is a larger distance between chromatid backbones ($\Delta R/R_b$ >10), and consequently, less overlap of chromatids ($V_o/V_o^{(0)} \approx 0.1$). The resulting linked chromatids are reminiscent of microscopy images of mitotic chromosomes (*Figure 2a*, right panel, and *e.g.*, [*Maeshima et al., 2005*]), as has been observed in previous simulations (*Goloborodko et al., 2016a*). Thus, we find that chromatin gaps left by pure one-sided extrusion inhibit the spatial resolution of linked chromosomes; moreover, determining the presence or lack of unlooped chromatin gaps in 1D is sufficient to predict the effects on 3D compaction. Together, these results indicate that while the two-sided loop extrusion model can explain condensin-mediated metazoan mitotic chromosome resolution, the pure one-sided loop extrusion model cannot.

## Semi-diffusive one-sided extrusion does not efficiently compact chromosomes

We next investigated the semi-diffusive one-sided extrusion model, in which the inactive LEF subunit may passively diffuse. We find that semi-diffusive LEFs can compact chromatin to a greater extent than pure one-sided LEFs in some scenarios, but are unable to achieve 1000-fold linear compaction for a plausible values of $\lambda/d$ (*i.e.*, $\lambda/d < 1000$, which is expected from experimental measurements (*Fukui and Uchiyama, 2007*; *Ganji et al., 2018*; *Gerlich et al., 2006a*; *Golfier et al., 2020*; *Kong et al., 2020*; *Takemoto et al., 2004*; *Terakawa et al., 2017*; *Walther et al., 2018*; *Figure 2d (i)*). The enhanced compaction by semi-diffusive one-sided LEFs arises from their ability to close some unlooped gaps (*Figure 2d (ii)*). LEFs may suppress gaps in two ways: 1) inactive but diffusive LEF subunits may stochastically slide toward each other and 2) diffusion of an inactive subunit of a 'parent' LEF may be rectified if a 'child' LEF is loaded within the loop so that the active subunit of the child LEF moves toward the inactive subunit of the parent LEF, leading to Brownian ratcheting (*Figure 2—figure supplement 3a*). The first mechanism is ineffective in eliminating gaps because it is opposed by the conformational entropy of the extruded loop (*Brackley et al., 2017*), and the LEFs may also diffuse apart, causing the unlooped gap to reappear. The second mechanism can be

enhanced by the active subunit of the child LEF actively 'pushing' the parent's inactive subunit (*Figure 2—figure supplement 4* and Appendix 1). These active processes are more effective at closing gaps. Nonetheless, Brownian ratcheting by nested LEFs does not sufficiently linearly compact chromosomes for all $\lambda/d < 1000$, while active pushing can only achieve a high degree of compaction if the active subunit can simultaneously reel chromatin through multiple inactive subunits and $\lambda/d \approx 1000$.

To understand how semi-diffusive LEFs enhance linear compaction in some particular scenarios, we investigated how compaction depends on the scaled diffusion speed, $v_{diff}/v$. For reference, with $v = 1$ kb/s as in vitro (*Ganji et al., 2018*; *Golfier et al., 2020*; *Kong et al., 2020*), $v_{diff}/v = 1$ corresponds to $D = 0.5$ kb$^2$/s or $D = 0.06$ µm$^2$/s on naked DNA, which is in the range of measured in vitro measured diffusion coefficients ($D = 0.01$–35 kb$^2$/s or 0.001–4 µm$^2$/s) for SMC complexes on DNA (*Davidson et al., 2016*; *Kanke et al., 2016*; *Kim and Loparo, 2016*; *Kim et al., 2019*; *Stigler et al., 2016*; *Terakawa et al., 2017*). For $v_{diff}/v <<1$, the inactive subunit diffuses very slowly, so the LEFs behave similarly to pure one-sided LEFs; moreover, thermal ratcheting by nested LEFs is very slow since the translocation speed of the active subunit of the child LEF is effectively limited by the diffusion of the inactive subunit of the parent LEF. Interestingly, in the case with rapid diffusion, $v_{diff}/v > 1$, semi-diffusive LEFs linearly compact chromosomes even less effectively than pure one-sided LEFs. Because conformational entropy favors shrinkage of parent loops, the diffusive subunit shrinks loops more rapidly than the active subunit grows loops. Since loops remain small, nesting of loops (i. e., LEFs extruding loops within loops) becomes less likely (*Figure 2—figure supplement 3*). Thus, gaps remain because they are not closed by Brownian ratcheting. Intriguingly, our simulations reveal that $v_{diff}/v \approx 1$ is an optimal case in which diffusion is sufficiently slow to permit loops to grow large enough to allow loop nesting, but fast enough to promote loop growth by thermal ratcheting. However, even this 'optimal' case leaves a large number of gaps. Thus, we find that for all $v_{diff}/v$ unlooped gaps remain (*Figure 2d (ii)*) and 1000-fold compaction cannot be achieved with $\lambda/d < 1000$ (*Figure 2d (i)*).

In the semi-diffusive model, as in the pure one-sided model, the limited ability to linearly compact chromosomes impairs 3D compaction. Simulated chromosomes are generally not rod-like (*Figure 2d (iii)*, inset), and the loop architecture remains gapped and weakly reinforced. Consequently, for optimal scaled diffusion speeds, $v_{diff}/v \approx 1$, the volume, $V$, is reduced by less than in the case of two-sided extrusion ($\leq 2$ fold vs. $>2.5$ fold, *Figure 2d (iii)*). Similarly, modest linear compaction of chromatids leads to only a slight increase in inter-chromatid distance (*Figure 2d (iv)*) and moderate overlap volume ($V_o/V_o^{(0)} \approx 0.2$). Thus, 3D compaction and sister chromatid resolution in the semi-diffusive model can exceed that of the pure-one sided model, but they still fall short of the far more dramatic compaction and distinct spatial resolution expected for mitotic chromosomes in vivo and reproduced by the two-sided loop extrusion model. The failure of this one-sided loop extrusion variant is again due to the inability to robustly eliminate unlooped gaps.

## One-sided loop extrusion with switching recapitulates mitotic compaction

The results of the previous sections suggest that robust mitotic chromosome compaction and chromatid resolution requires LEFs that consistently and irreversibly eliminate unlooped gaps. We therefore consider a variation of the one-sided extrusion model in which only one LEF subunit translocates at a time, but the LEFs stochastically switch which subunit is active at rate $k_{switch}$. In principle, in this scenario, LEFs may be 'effectively two-sided,' which allows LEFs initially in a divergent orientation ($\leftarrow \rightarrow$) to eliminate the initially unlooped gap (*Banigan and Mirny, 2019*).

To study mitotic chromosome compaction within the switching model, we vary both $\lambda/d$ and the scaled switching rate, $k_{switch}/k_{unbind}$. The scaled switching rate determines the number of times that a LEF will switch before unbinding; each switch allows a LEF the chance to close a gap (*Banigan and Mirny, 2019*). Accordingly, we observe that the ability of LEFs to linearly compact chromatin increases with $k_{switch}/k_{unbind}$. For very slow switching rates ($k_{switch}/k_{unbind} <<1$, or roughly $k_{switch} <<1$ min$^{-1}$ for experimentally observed $k_{unbind}$ [*Ganji et al., 2018*; *Gerlich et al., 2006a*; *Terakawa et al., 2017*; *Walther et al., 2018*]), loop extrusion is effectively one-sided because switches rarely occur and gaps are not closed, so linear compaction is limited to ~10 fold (*Figure 2e (i), (ii)*, blue). For faster scaled switching rates ($0.1 < k_{switch}/k_{unbind} \leq 1$), switches are more likely to occur during each LEF's residence time, so greater numbers of LEFs are effectively two-sided and

more gaps can be closed (*Figure 2e (i), (ii)*, gray). In these cases, LEFs linearly compact chromosomes 10- to 100-fold. For very fast switching ($k_{switch}/k_{unbind} > 1$ or $k_{switch} > 1$ min$^{-1}$), many switches occur per residence time. Thus, all LEFs are effectively two-sided so that all unlooped gaps are eliminated for large $\lambda/d$, and 1000-fold linear compaction can be achieved (*Figure 2e (i), (ii)*, red).

Concordant with observations for linear compaction, we find that 3D chromosome compaction and resolution varies from the one-sided to two-sided phenotypes with increasing scaled switching rate, $k_{switch}/k_{unbind}$. Chromosomes with rapidly switching LEFs can undergo a large reduction in volume, $V$ (>2.5 fold, *Figure 2e (iii)*), comparable to what is observed for two-sided extrusion. Similarly, sister chromatid resolution can be achieved in the switching model for $k_{switch}/k_{unbind} > 1$. The distance between chromatid backbones increases ($\Delta R/R_b > 8$, *Figure 2e (iv)*), and overlap is greatly reduced ($V_o/V_o^{(0)} \approx 0.1$), comparable to what is achieved in the two-sided model. We thus conclude that the switching model with fast switching rates, $k_{switch} \sim 1$ min$^{-1}$, can reproduce the experimentally observed 3D compaction and resolution of mammalian mitotic chromosomes.

Of the three main variants of one-sided loop extrusion that we tested, only the switching model can reproduce mammalian mitotic chromosome compaction and resolution. In each of these models, the ability of LEFs to eliminate unlooped gaps governs compaction and resolution. Chromatin segments that are not linearly compacted into loops are longer, and thus have a larger 3D size. Therefore, the average number of unlooped gaps that remain, a 1D quantity, determines the 3D structure and organization of simulated mitotic chromosomes. Effectively two-sided extrusion is required to eliminate these gaps, and of the models considered here, this physical mechanism is reliably present in only the switching model.

## Attractive interactions between LEFs cannot rescue one-sided extrusion

As an alternative to the models above, which are dominated by the effects of extrusion-driven linear compaction, we performed polymer simulations to determine whether gaps created by one-sided loop extrusion could be eliminated by 3D attractive interactions between LEFs or between different polymer segments (*e.g.*, poor solvent). Moreover, we explored whether such interactions could volumetrically compact chromosomes and generate rod-like mitotic chromosomes, as previously suggested (*Sakai et al., 2018*). We find that 3D attractions can volumetrically compact polymers (*Figure 2—figure supplement 5a*), but the resulting structures do not resemble mitotic chromosomes. When LEFs attract each other, compacted chromosomes form extended, clumpy structures (*Figure 2—figure supplement 5b*, top), and chromatin gaps remain visible. Moreover, sister chromatids do not spatially segregate (*Figure 2—figure supplement 5b*, bottom). When the simulated chromosomes are instead treated as polymers in poor solvent, chromosomes are compacted into spherical structures and sister chromatids cannot be spatially resolved (*Figure 2—figure supplement 5c*). Attractive interactions have little effect on chromosome structure when the interaction strength, $\varepsilon$, is low, but when $\varepsilon$ is large, the chromosome is compacted into a spherical globule. These findings are consistent with previous theoretical and computational work on polymer combs (*Fytas and Theodorakis, 2013*; *Sheiko et al., 2004*), showing that 3D attractive interactions lead to a coil-globule transition.

We also considered the possibility that interactions between one-sided LEFs might alter their residence times. We hypothesized that such interactions could stabilize LEFs that had closed gaps. However, we found that linear compaction in this model is still limited to 10-fold because gaps are still created by divergently extruding LEFs (*Figure 2—figure supplement 6*). Altogether, we find that attractive interactions between LEFs or between different polymer segments cannot be the mechanism of gap closure for mitotic chromosomes.

## LEF traversal might rescue one-sided extrusion

Recent single-molecule experiments report the first observations of effectively two-sided loop extrusion that results from the coordinated activity of two one-sided loop extruders (*Kim et al., 2020*). Single-molecule experiments have shown that yeast condensins can form 'Z-loops' that act as an effectively two-sided extruder. In this scenario, condensins can pass each other as they translocate along DNA, thus forming structures that reel in DNA from two directions. To analyze this possibility, we simulated chromosomes compacted by LEFs that can freely traverse each other. In this model, linear chromosome compaction, as quantified by loop coverage, increases exponentially with $\lambda/d$, as

expected from theory (*Figure 2f* and Appendix 2). Correspondingly, we observe that chromosomes in this model form compact, rod-like structures (*Figure 2g*). We find that ~ 1000 fold linear compaction is achieved for $\lambda/d \sim 7$, which can be satisfied with reasonable physiological values of loop sizes, $\ell = \lambda \sim 140$ kb (*Earnshaw and Laemmli, 1983*; *Gibcus et al., 2018*; *Naumova et al., 2013*; *Paulson and Laemmli, 1977*) and densities of one LEF per $d \sim 20$ kb (*Fukui and Uchiyama, 2007*; *Takemoto et al., 2004*; *Walther et al., 2018*). In addition, LEFs in this model can spatially resolve sister chromatids (*Figure 2h*). Thus, one-sided LEFs that can freely traverse each other may be sufficient to compact and resolve mitotic chromosomes.

## Formation of interphase chromosome TADs, stripes, and dots
### Model and observables
Next, we determined whether one-sided extrusion can recapitulate prototypical features in Hi-C and micro-C maps (*Krietenstein et al., 2020*) of vertebrate cells during interphase, such as TADs, 'stripes' (also called 'lines,' 'tracks,' or 'flames'), and particularly, the 'dots' (or 'corner peaks') found at the boundaries of TADs (*Figure 3a*). Dots are foci on Hi-C maps that reflect enriched contact frequency between specific loci, often found at the corners of TADs and/or between proximal (<1–2 Mb) CTCF sites (*Krietenstein et al., 2020*; *Rao et al., 2014*). TADs, stripes, and dots are cohesin-mediated, and they can be modulated by changes to cohesin and/or CTCF. Thus, we evaluate extrusion models based on whether they can generate these hallmarks of interphase chromosome organization.

We perform polymer simulations for each model, sweeping $\lambda$ and $d$ (*Cattoglio et al., 2019*; *Fudenberg et al., 2016*; *Holzmann et al., 2019*), as well as model-specific parameters. CTCF barriers are modeled as partially permeable loop-extrusion barriers (*Fudenberg et al., 2016*; *Nuebler et al., 2018*). In *Figure 3* we use experimental values for $\lambda$ and $d$ for wild-type (WT) conditions (Materials and methods and *Figure 3—figure supplement 1*); other values for $\lambda$ and $d$ are explored in the figure supplements. We compute and visualize contact maps from these simulations and quantify the dot strength by the enhancement of dot contact frequency over background, as in *Figure 3—figure supplement 2*; *Gassler et al., 2017*).

### Pure one-sided extrusion can reproduce some but not all features of interphase organization
In models of two-sided loop extrusion in interphase, a TAD arises due to the formation of extruded loops within a particular region, usually bounded by convergently oriented CTCF sites. A stripe emerges if one extruding subunit of a LEF is stalled by CTCF while the other subunit continues extruding (*Figure 3—figure supplement 3*). A dot arises when two barriers to extrusion (*e.g.*, convergently oriented CTCF sites) are brought together by one or a few LEFs that close a gap between two barriers (*Figure 3—figure supplement 3*; *Fudenberg et al., 2016*; *Sanborn et al., 2015*).

While two-sided extrusion can reproduce TADs, stripes, and dots, we found that the simplest model of one-sided extrusion can recapitulate only some of these features. When LEFs are uniformly loaded onto chromatin, pure one-sided extrusion can form the bodies of TADs and stripes, but does not form dots (*Figure 3b*, right panel). For one-sided extrusion, stripes are an average effect of LEFs loading at different loci and extruding up to a barrier (*Figure 3—figure supplement 3*), while dots are not formed because only one-sided LEFs loaded at a barrier can pair two barriers (*Figure 3—figure supplement 3*). This problem cannot be resolved by increasing the processivity, $\lambda$, or decreasing the separation between LEFs, $d$ (*Figure 3—figure supplement 4*). In contrast, two-sided extrusion with increased processivity generates the strong dots seen in wild-type data as well as the 'extended dots' (*Figure 3b* and *Figure 3—figure supplement 5*) seen in Wapl depletion data (*Gassler et al., 2017*; *Haarhuis et al., 2017*; *Wutz et al., 2017*). This failure to form dots is due to inevitable gaps that one-sided extrusion leaves between LEFs and between LEFs and CTCF barriers (*Figure 3c*).

### Semi-diffusive one-sided extrusion cannot produce Hi-C dots
The semi-diffusive model creates a phenotype that is similar to that of pure one-sided extrusion for simulations of WT conditions (*Figure 3d*); it can generate TAD bodies and stripes, but neither dots nor extended dots (*Figure 3—figure supplement 2*). We conclude that the semi-diffusive one-sided

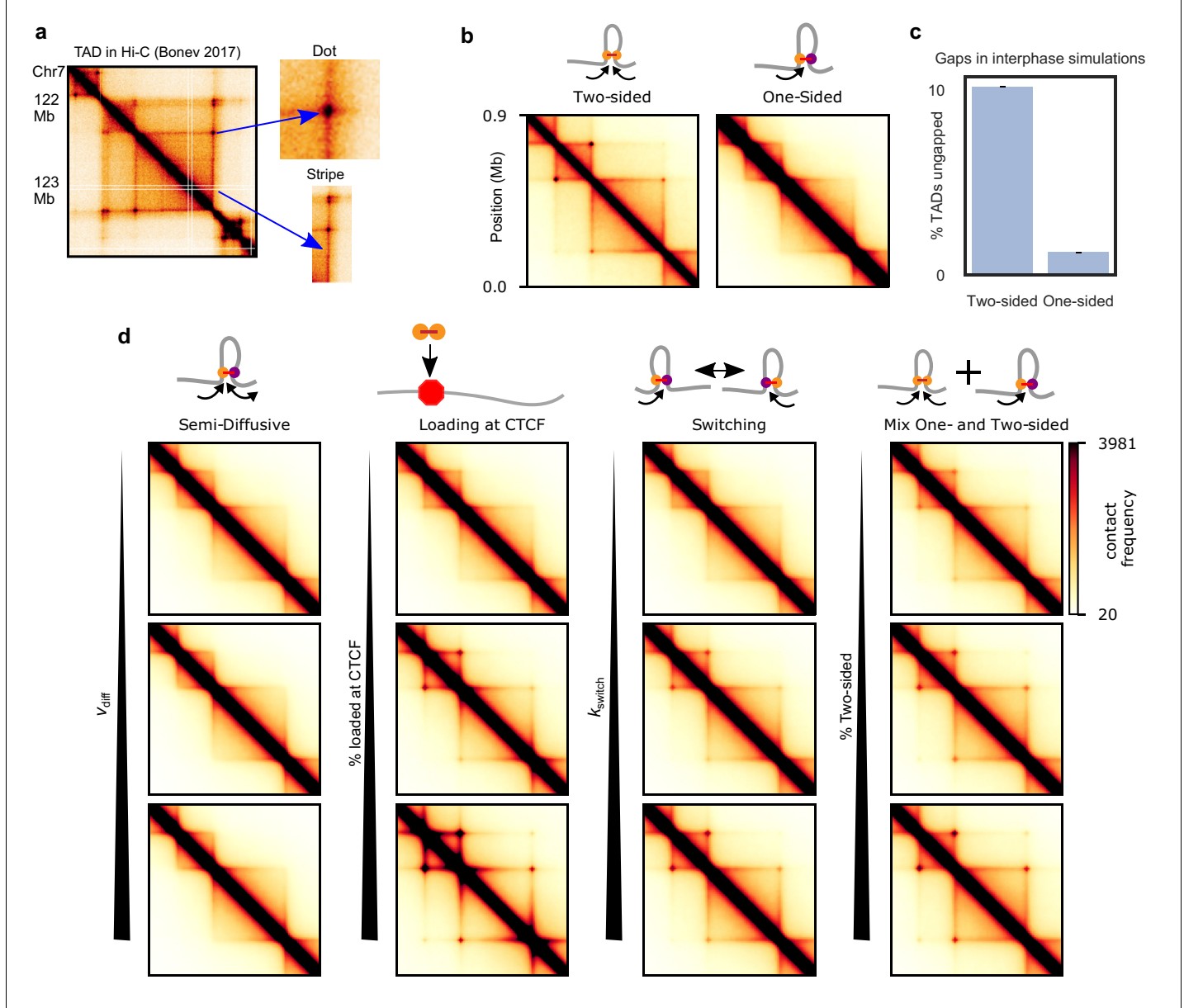

**Figure 3.** TADs and corner peaks for variations on one-sided loop extrusion. (**a**) A TAD in Hi-C of cortical neurons (*Bonev et al., 2017*), visualized by HiGlass (*Kerpedjiev et al., 2018*) at a resolution of 8 kb. Two characteristic features of TADs, stripes and dots, are indicated. (**b**) Contact maps computed from polymer simulations with two-sided (left) and one-sided (right) LEFs. The residence time and density of LEFs have been chosen to approximate the WT conditions ($d=\lambda=200$ kb) (Materials and methods and *Figure 3—figure supplement 1*). (**c**) Percentage of ungapped TADs for the same LEF separation and processivity as in (**b**). The percentage of ungapped TADs is computed over 100,000 LEF turnover times, for a system of 20 TADs of size 400 kb, the same size as the largest TAD in the contact maps. The standard error in the mean of the percentage of ungapped TADs is less than 0.05%. (**d**) Contact maps computed from polymer configurations for the semi-diffusive model, the one-sided model with biased loading, the switching model, and the model with a mix of one- and two-sided LEFs. WT values of $d$ and $\lambda$ are used for every map. The parameter values, from top to bottom and from left to right, are: $v_{diff}/v = 0.1$, 1, and 3.5 (with $v = 1$ kb/s, $D = 0.2$, 2, and 7 kb$^2$/s), bias for loading at CTCF = 10, 100, and 1000, $k_{switch}/k_{unbind} = 0.1$, 1, and 10 and percentage two-sided = 20, 40, and 60.

The online version of this article includes the following figure supplement(s) for figure 3:

**Figure supplement 1.** Comparison of the contact probability as a function of genomic separation (scalings) of experiments (*Haarhuis et al., 2017*) and simulations to validate the chosen parameters for the simulations.

**Figure supplement 2.** The primary and extended dot strength for two-sided, one-sided, semi-diffusive and switching LEFs.

**Figure supplement 3.** Illustrations of loop extrusion by one-sided and two-sided LEFs.

**Figure supplement 4.** Sweep of the separation between LEFs and the processivity of LEFs for one-sided LEFs.

*Figure 3 continued on next page*

*Figure 3 continued*

**Figure supplement 5.** Sweep of the separation and the processivity of LEFs for two-sided LEFs.
**Figure supplement 6.** Sweep of the separation and the processivity of LEFs for one-sided LEFs with a loading bias at CTCF sites.
**Figure supplement 7.** Sweep of the separation and the processivity of LEFs for one-sided LEFs that may traverse each other.
**Figure supplement 8.** Illustration of how the moving barrier mechanism (*Brandão et al., 2019*) combined with one-sided LEFs may result in dots in Hi-C of *S. cerevisiae*.

model works similarly to the pure one-sided model, and it is also limited by its inability to close gaps between LEFs and between LEFs and barriers.

## One-sided extrusion with preferential loading at TAD boundaries

Next, we considered variations of the model in which one-sided LEFs are loaded nonuniformly, with increased probability of loading at barriers (*Nichols and Corces, 2015*; *Rubio et al., 2008*; *Figure 3d*). Each barrier has two loading sites and one-sided LEFs are loaded directionally so that they translocate away from the boundary. Loading of LEFs at CTCF sites increases both the primary and extended dot strengths, qualitatively reproducing both wild-type conditions ($\lambda$ = 200 kb, $d$ = 200 kb) (*Figure 3d*) and Wapl depletion ($\lambda$ = 2 Mb, $d$ = 200 kb) conditions (*Figure 3—figure supplement 6*). To clearly observe dots, however, LEFs must have a strong loading bias, *i.e.*, >100 fold preference to bind barrier sites as compared to body sites. While contacts within the TAD body are reduced for this large bias (*Figure 3d*), it is possible to find a loading bias and LEF density such that both dots and the TAD body are clearly visible (*Figure 3—figure supplement 6*). Although current experimental evidence does not support preferential loading of cohesin at CTCF sites in mammals (*Busslinger et al., 2017*; *Fudenberg et al., 2017*; *Nora et al., 2019*; *Nora et al., 2017*; *Parelho et al., 2008*; *Wendt et al., 2008*), such a mechanism of TAD, stripe, and dot formation is feasible and may be operational under some conditions, in some cell types, or in other species.

## One-sided extrusion with switching reproduces all features of interphase organization

We hypothesized that mechanisms other than loading at CTCF could enable one-sided extrusion to reproduce interphase Hi-C features. We considered the switching model because a LEF, when switching frequently enough, might bring two barriers together, even if it is not loaded at a barrier. Moreover, switching could eliminate gaps between nearby LEFs.

The switching model for slow switching rates approximates the pure one-sided model; primary and extended dots are not present (*Figure 3d*, third column) and they do not appear with increased $\lambda$ (*Figure 3—figure supplement 2*). For faster switching rates, primary and extended dots appear (and loop strengths increase with $\lambda$, *Figure 3—figure supplement 2*), as they do in the two-sided model (*Figure 3d*, third column). The switching model approaches the two-sided extrusion model, as quantified by primary and extended dot strengths for $k_{switch}/k_{unbind} \approx 10$ (*Figure 3—figure supplement 2*). Thus, the model suggests that cohesin must undergo a switch once per minute for characteristic residence times of ~10–20 min (*Gerlich et al., 2006b*; *Hansen et al., 2017*; *Kueng et al., 2006*; *Stigler et al., 2016*; *Tedeschi et al., 2013*; *Wutz et al., 2017*). In addition to dots, switching generates a high frequency of intra-TAD contacts and stripes (*Figure 3d*, third column). Thus, one-sided LEFs that switch sufficiently fast can account for features of interphase chromosome organization.

## A mix of one- and two-sided extrusion can reproduce features of interphase organization

A mix of one- and two-sided LEFs approaches either the one-sided or the two-sided phenotype depending on the percentage of two-sided LEFs (*Figure 3d*, right column). Dots are visible, but weak for a mix with 20% two-sided LEFs, while a mix with 60% two-sided LEFs approaches the two-sided dot strength and generates stripes and intra-TAD contacts (*Figure 3d*, right column). A lower percentage of two-sided extruders, however, is needed to reproduce interphase organization (~50%) as compared to the percentage needed for strong mitotic compaction (>80%). While even a

small fraction of gaps can be detrimental to mitotic compaction, gaps between LEFs are less damaging for the interphase, in which LEFs are more sparse along the chromosome (*Figure 3c*).

## LEF traversal might rescue one-sided extrusion for small enough LEF separations

Next, we considered one-sided LEFs that may traverse each other upon encountering each other as a model for 'Z-loops,' which have been observed for yeast condensins on DNA (*Kim et al., 2020*). We find that under WT conditions ($d=\lambda=200$ kb), such LEFs do not form noticeable dots (*Figure 3—figure supplement 7*). While the ability of LEFs to traverse each other can eliminate both gaps between LEFs and gaps between LEFs and boundaries, one-sided extruders with LEF traversal are still less efficient in pairing CTCF sites than two-sided LEFs. Dots become stronger when the separation between LEFs is reduced ($d \leq 50$ kb) while maintaining the WT processivity or the processivity is increased ($\lambda >2$ Mb) while maintaining WT LEF densities for the simulated TAD sizes. Nonetheless, dots remain weaker than those of two-sided LEFs with the same separation and processivity.

Our simulations show that features of interphase chromosome organization can be reproduced by variants of one-sided extrusion where (a) extruders can switch their directionality approximately every minute; (b) one-sided extruders are mixed with two-sided extruders; (c) extruders have a > 100 fold preference for loading at CTCF sites; or (d) extruders may traverse each other and have a small average separation ($d \leq 50$ kb) or large processivity ($\lambda >2$ Mb).

## Juxtaposition of bacterial chromosome arms

### Model and observables

The bacterial SMC complex (bSMC) plays a direct role in juxtaposing the arms of the circular bacterial chromosome. In bacteria such as *B. subtilis*, the strong site-specific loading of bSMC followed by loop extrusion forms a distinctive pattern (*Minnen et al., 2016*; *Tran et al., 2017*; *Wang et al., 2017*) different from the case of uniform loading (assumed for eukaryotic systems). The bSMC loading sites (*i.e., parS* sites) are typically located near the origin of replication (<100 kb away). A secondary diagonal is visible emanating from the *parS* site in the bacterial Hi-C maps; it indicates long-ranged, high frequency contacts between chromosomal loci on opposite sides of the replichore (*Figure 4a*; *Le et al., 2013*; *Marbouty et al., 2015*; *Wang et al., 2015*). This secondary diagonal arises due to the high processivity of bSMCs ($\lambda >4$ Mb), which brings together DNA segments approximately equidistant from the origin-proximal *parS* loading sites. Recent modeling studies show that the shape and trajectory of the secondary diagonal can be theoretically predicted by a stochastic model of bSMC two-sided loop extrusion (*Brandão et al., 2019*; *Miermans and Broedersz, 2018*). In light of these recent models and data, we explore the extent to which variations of one-sided extrusion might recapitulate these results.

We compare the models for one-sided extrusion as follows. We perform 1D simulations of LEF dynamics, and then use our semi-analytical approach (see Materials and methods and Appendix 3) to produce Hi-C-like contact maps. In contrast to the previous sections, we only consider the limit of large $\lambda/d > 1$ as suggested by experiments (*i.e., d* < 4 Mb < $\lambda$; see Appendix 3; *Tran et al., 2017*; *Wang et al., 2017*; *Wilhelm et al., 2015*). We evaluate the model by qualitatively comparing the width, intensity, and length of the experimental secondary diagonals to what is produced by our models.

### Pure one-sided extrusion does not produce symmetric arm juxtaposition

It was recently shown by 3D polymer simulations that the pure one-sided loop extrusion model cannot reproduce the secondary diagonals visible by Hi-C (*Miermans and Broedersz, 2018*). In contrast, two-sided loop extrusion qualitatively reproduced the experimentally observed secondary diagonal (*Miermans and Broedersz, 2018*), with an intensity that depends on the number of LEFs (*Figure 4—figure supplement 1*, left column).

Using our semi-analytical approach, we recapitulate these previous results (*Figure 4b*) and explore a broader range of parameter values. As seen in *Figure 4b* (right panel), with bSMC loading only at a predetermined site (with up to 30 bSMCs per origin of replication [*Graham et al., 2014*; *Wilhelm et al., 2015*]), one-sided extrusion fails to yield the secondary diagonal that is characteristic of the chromosome contact maps of *B. subtilis* (*Figure 4a*) and other bacteria (*Böhm et al., 2020*;

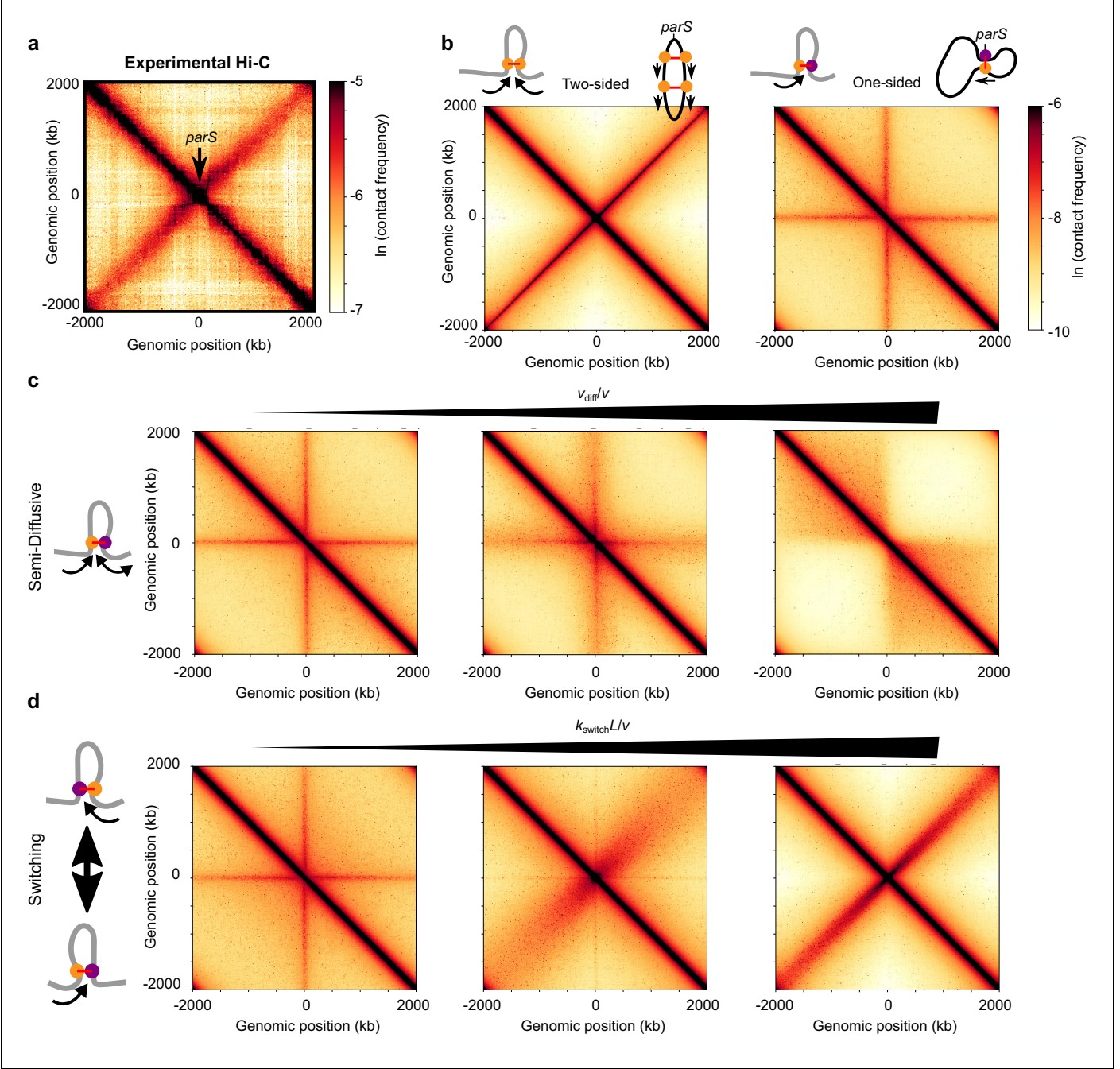

**Figure 4.** Effect of different extrusion rules on bacterial contact maps. (**a**) Experimental Hi-C map for *B. subtilis* with a single *parS* site (SMC complex loading site) near the *ori* in the strain BDR2996 from *Wang et al. (2015)*. Simulations of (**b**) the pure two-sided model (left map, and schematic of a single two-sided LEF and a chromosome extruded by two-sided LEFs) and the pure one-sided model (right map and schematic). (**c**) Simulations of the semi-diffusive model (with diffusive stepping rates, from left to right, of $v_{diff}/v$ = 0.005, 0.1, and 3.5 ($D$ = 0.005, 0.1, and 3.5 kb$^2$/s with $v$ = 1 kb/s)), and (**d**) the switching model (with switching rates, from left to right, of $k_{switch}L/v$ = 4, 40, and 400, or $k_{switch}$ = 0.001, 0.01, and 0.1 s$^{-1}$, respectively). All simulations displayed were performed with $N$ = 5 LEFs per chromosome.

The online version of this article includes the following figure supplement(s) for figure 4:

**Figure supplement 1.** Contact maps from simulations for different mixes of one- and two-sided LEFs and numbers of LEFs for bacterial chromosomes.
**Figure supplement 2.** Contact maps from simulations for different values of the LEF stepping probability (per simulation step), with $N$ = 5 LEFs on each chromosome.
**Figure supplement 3.** Sweep of the diffusive stepping rate and the number of LEFs for bacterial chromosomes.

*Figure 4 continued on next page*

*Figure 4 continued*

**Figure supplement 4.** Contact maps from 3D polymer simulations of an extrusion model in which LEFs may traverse each other and may occupy the same lattice sites.

**Figure supplement 5.** Contact maps from simulations for scaled switching rates and numbers of LEFs for bacterial chromosomes.

**Figure supplement 6.** Contact maps generated from molecular dynamics simulations as compared to the semi-analytical method.

**Figure supplement 7.** Contact probability as a function of genomic distance generated from molecular dynamics simulations as compared to the semi-analytical method.

**Figure supplement 8.** Generating Gaussian chain contact maps analytically from loop configurations.

*Le et al., 2013*; *Marbouty et al., 2014*; *Umbarger et al., 2011*; *Wang et al., 2015*). Instead, pure one-sided extrusion exhibits a '+"-shaped pattern overlaid on the main diagonal, which indicates contacts of the *parS* loading site with all other chromosomal loci. This results from the fact that in pure one-sided loop extrusion, one LEF subunit is fixed at the *parS* loading site, while the other subunit translocates away from it. Thus, we conclude that pure one-sided loop extrusion fails to reproduce the symmetric chromosome arm juxtaposition that is characteristic of many bacterial Hi-C maps.

## Semi-diffusive one-sided extrusion does not properly juxtapose chromosome arms

We next considered the semi-diffusive case in which one subunit of the LEF actively translocates, while the other diffuses. Despite the increased mobility of the inactive subunit, the qualitative patterns of the contact map remained largely unchanged from the pure one-sided model (*Figure 4c*). Increasing the scaled subunit diffusion rate, $v_{diff}/v$, broadened the '+"-shaped pattern and did not produce the secondary diagonal (*Figure 4c* and *Figure 4—figure supplement 2*). Interestingly, for high enough values of $v_{diff}/v$ (*Figure 4c*, right panel), the '+"-shaped pattern is replaced by a square TAD-like structure, reminiscent of two large macrodomains separating each of the sister replichores from each other. No secondary diagonal was observed even when the number of LEFs that is present on the chromosome is changed (*Figure 4—figure supplement 3*). Thus, for all values of $v_{diff}/v$, the semi-diffusive loop-extrusion model does not explain the available Hi-C data for *B. subtilis* and *C. crescentus* (and other bacteria with a secondary diagonal).

## One-sided extrusion with LEF traversal does not properly juxtapose chromosome arms

We also tested whether one-sided loop extrusion with traversal could explain the experimental data. Similarly to the semi-diffusive case in which $v_{diff}/v$ is large (*Figure 4c*, right panel), we found that LEF traversal generated a square, TAD-like structure between the left and right replichores (*Figure 4—figure supplement 4*), rather than a secondary diagonal characteristic of prokaryotes with an SMC/ *parABS* system.

## One-sided extrusion with directional switching can juxtapose chromosome arms

We next tested whether one-sided LEFs that stochastically switch which subunit is active can recapitulate the available data. We performed a parameter sweep over a range of numbers of bSMCs and scaled switching rates, $k_{switch}L/v$, and we generated Hi-C contact maps (*Figure 4d* and *Figure 4— figure supplement 5*). The width of the experimentally observed secondary diagonal constrains the possible values of $k_{switch}L/v$ in our model. In experiments, the secondary diagonal is narrow, with a width of ~100 kb across the entire map. This suggests that there is very little variance in the extrusion speeds along each chromosome arm. With more frequent switches (larger $k_{switch}L/v$), the progression of each extruding subunit along each arm varies less relative to the mean extrusion trajectory (*Figure 4d*). We found that fast enough switching rates ($k_{switch}L/v > 200$) can produce the secondary diagonal (*Figure 4d*), irrespective of the number of bSMCs (*Figure 4—figure supplement 5*). For *B. subtilis* and *C. crescentus*, we calculate that the upper bound on the mean time between switches is approximately 2–10 s and 10–20 s, respectively, with $v$ = 50 kb/min in *B. subtilis*

and *v* = 25 kb/min in *C. crescentus* as measured experimentally (*Figure 4d*, right panel) (*Tran et al., 2017*; *Wang et al., 2017*).

Thus, in contrast to other models that we considered, one-sided extrusion with switching can juxtapose chromosomal arms, as demonstrated by the presence of the Hi-C secondary diagonal that is prominent in many bacterial maps. In our model, this requires a relatively fast switching rate, which effectively makes a one-sided LEF behave like a two-sided LEF at the physiologically relevant time scales of a few minutes. Other variants of one-sided mechanism cannot achieve juxtaposition of bacterial arms due to tethers that remain between distal chromosome loci and the LEF loading site, indicating that bSMC is an effectively two-sided extruder.

## Discussion

SMC complexes are ubiquitously found in all domains of life, and strong evidence is emerging that SMC protein complexes function by DNA loop extrusion, which appears to be central to their function. By forming loops, SMC complexes promote chromosome contacts spanning tens of kilobases to megabases in bacteria (*Le et al., 2013*; *Lioy et al., 2018*; *Marbouty et al., 2015*; *Wang et al., 2015*) and hundreds of kilobases in metazoan cells (*e.g.,* [*Busslinger et al., 2017*; *Gassler et al., 2017*; *Gibcus et al., 2018*; *Rao et al., 2017*; *Rao et al., 2014*; *Schwarzer et al., 2017*; *Wutz et al., 2017*]). Proper function of the SMC machinery is vital to chromosome organization and compaction. Improper chromosome compaction and segregation can lead to anaphase bridges in metazoan cells (*Charbin et al., 2014*; *Green et al., 2012*; *Hagstrom et al., 2002*; *Nagasaka et al., 2016*; *Piskadlo et al., 2017*; *Steffensen et al., 2001*) and mispositioning of origins of replication in prokaryotes (*Wang et al., 2014*), all of which might cause aneuploidy (or anucleate cells in bacteria) and DNA damage (*e.g.,* [*Fenech et al., 2011*; *Martin et al., 2016*; *Wang et al., 2013*]). Additionally, the loss of interphase chromosome structure in vertebrates by loss of cohesin SMC complexes can affect gene expression (*e.g.,* [*Bompadre and Andrey, 2019*; *Cuartero et al., 2018*; *Delaneau et al., 2019*; *Lupiáñez et al., 2015*; *Merkenschlager and Nora, 2016*; *Nora et al., 2017*; *Rao et al., 2017*; *Schoenfelder and Fraser, 2019*; *Schwarzer et al., 2017*; *Seitan et al., 2013*]). Similarly, mutations that perturb cohesin or condensin can lead to human developmental disorders, such as Cornelia de Lange syndrome (*de Lange, 1933*) and microcephaly (*Martin et al., 2016*).

Recent in vitro imaging studies showed that loop extrusion by *Saccharomyces cerevisiae* condensin SMC complexes is purely one-sided (*Ganji et al., 2018*). To determine the biophysical implications and to test the generality of this striking molecular observation, we explored whether one-sided loop extrusion could explain SMC-dependent phenomena observed in vivo for a range of organisms beyond *S. cerevisiae*. These phenomena included mitotic chromosome compaction in metazoans, formation of TADs and dots (corner peaks) in vertebrate interphase Hi-C maps, and juxtaposition of chromosome arms in rapidly growing bacteria. Together, these three systems exhibit the main features of chromosome organization that are attributed to loop extrusion: linear and 3D compaction, spatial segregation, *cis* loop/domain formation, linear scanning in cis, and progressive juxtaposition of chromatin flanking a loading site.

Our work, along with recent theoretical modelling (*Banigan and Mirny, 2019*; *Miermans and Broedersz, 2018*), indicates that pure one-sided loop extrusion does not generically reproduce these three phenomena, except under specific conditions. Therefore, biophysical capabilities beyond the one-sided loop extrusion observed for yeast condensins in vitro should be present for other organisms. Indeed, recent experimental evidence suggests that pairs of yeast condensins may be able to cooperatively grow loops bidirectionally (*Kim et al., 2020*), while human and *Xenopus* condensins and cohesins can perform either one- or two-sided loop extrusion (*Davidson et al., 2019*; *Golfier et al., 2020*; *Kim et al., 2019*; *Kong et al., 2020*; *Moevus, 2019*). Thus, we explored simple variations of the pure one-sided loop extrusion model and identified a class of one-sided extrusion models that can reproduce in vivo experimental observations (*Table 1*). Our results suggest modes of loop extrusion that might be observed in future experiments.

### A framework for modeling SMC complex dynamics

We focused on several variations of the one-sided loop extrusion model and investigated the consequences for 3D chromosome organization (*Table 1*). Our aim was not to exhaustively enumerate all possible model variations of one-sided extrusion. Instead, we sought to obtain and evaluate a set of

**Table 1.** Summary of model results.

Each entry indicates whether there are parameters for the specified model (column headings) that can explain chromosome organization in the specified scenario (row headings). A dash indicates that the model/scenario combination was not explored. *Indicates theoretical result from *Banigan and Mirny, 2019*.

| | Pure 1-sided | 2- sided | 1-sided + 2-sided mix | Semi- diffusive | 1-sided + loading bias | Switching | 1-sided with traversal | 1-sided + 3D attraction |
|---|---|---|---|---|---|---|---|---|
| Mitosis | No | Yes | Yes with > 80% 2-sided | No | Yes with > 1000 fold bias* | Yes with $k_{switch}/k_{unbind} > 10$ | Yes | No |
| Interphase | No | Yes | Yes with > 50% 2-sided | No | Yes with > 100 fold bias | Yes with $k_{switch}/k_{unbind} > 10$ | Yes for $d \leq 50$ kb or $\lambda > 2$ Mb | No** |
| Bacteria | No | Yes | No | No | No | Yes with $k_{switch}L/v > 200$ | No | - |

**Indicates inferred from simulation results of *Fudenberg et al., 2016*.

minimalistic requirements to explain experimental data. We modeled SMC complexes as LEFs with two subunits with distinct dynamics; subunits could be either active (*i.e.*, moving processively), inactive and anchored, or inactive but diffusive. Within this framework of varying the dynamics of the subunits, we primarily focused on the following models for LEFs: 1) one subunit active, the other subunit inactive and anchored ('pure one-sided'), 2) one subunit active, the other subunit inactive but diffusive ('semi-diffusive'), 3) one subunit active, the other subunit anchored, with kinetic interchange of active and anchored subunits ('switching'). We also considered several related variants for each chromosome organization scenario, such as preferential loading at CTCF by one-sided cohesins during interphase. As a point for comparison, we quantitatively compared all results with those of two-sided extrusion, which previous works have shown to recapitulate key experimental observations (*Alipour and Marko, 2012*; *Brandão et al., 2019*; *Fudenberg et al., 2016*; *Goloborodko et al., 2016a*; *Goloborodko et al., 2016b*; *Miermans and Broedersz, 2018*; *Sanborn et al., 2015*).

## Unlooped chromatin from one-sided extrusion hinders chromosome compaction and organization for higher eukaryotes

Our modeling demonstrates that the ability to robustly eliminate unlooped gaps is essential to the chromosome-organizing role of LEFs. As a result, models in which gaps persist in steady state, such as the pure one-sided model, fail to reproduce hallmarks of chromosome organization found in several physiological scenarios. One-sided extrusion generally does not reproduce mitotic chromosome compaction and chromatid segregation or hallmarks of interphase Hi-C maps, without further assumptions beyond what has been observed experimentally. Importantly, even dynamic LEF turnover (*i.e.*, allowing dynamic chromatin unbinding with uniform rebinding) does not eliminate gaps because LEF unbinding (and even LEF binding) can introduce new gaps. Instead, chromosome compaction, resolution, and interphase organization can readily be explained by physical mechanisms that either eliminate gaps by turning one-sided extrusion into effectively two-sided extrusion (*e.g.*, as in the switching model) or suppress the creation of gaps (*e.g.*, by biased loading at boundaries).

In the case of mitotic chromosome compaction, linear compaction by pure one-sided loop extrusion is limited to ~10 fold because it unavoidably leaves gaps between SMC complexes (*Figure 2c (i), (ii)* and [*Banigan and Mirny, 2019*]). By simulations, we showed that 10-fold linear compaction is not sufficient to reproduce the classical 3D shapes of mitotic chromatids and chromosomes are volumetrically compacted at most twofold in 3D (*Figure 2 b,c (iii)*). This defect in 3D compaction leads to defects in mitotic chromosome resolution (*Figure 2 b,c (iv)*). Allowing the SMC complexes' anchor points to diffuse (*i.e.*, slide) along chromosomes also does not close gaps because loop formation is opposed by the conformational entropy of the formed loop (*Figure 2d (ii)* and *Figure 2—figure supplement 3*). Therefore, the LEFs cannot generate a sufficient increase in linear compaction for any diffusive stepping rate, $v_{diff}$ (or diffusion coefficient, $D$) (*Figure 2d (i)*); in vitro experiments also show that one-sided condensins with diffusing safety belts do not grow large DNA loops (*Ganji et al., 2018*). More generally, with one-sided LEFs, uncompacted gaps are pervasive, so simply adding a small fraction of two-sided LEFs is unable to sufficiently compact chromosomes; in vivo levels of compaction requires >80% two-sided LEFs (*Figure 2—figure supplement 1*; *Banigan and Mirny, 2019*). Similarly, a model in which LEFs are effectively two-sided, such as the switching model

in which the active and inactive subunits dynamically switch, can generate greater than twofold 3D compaction and clear resolution of sister chromatids (*Figure 2e (iii), (iv)*), as observed in vivo. Such a switching mechanism could be achieved in vivo by a stochastic strand switching mechanism in which both upstream and downstream DNA can be captured by the loop extruder (*Hassler et al., 2018*; *Marko et al., 2019*).

For interphase organization in vertebrate cells, the ability of one-sided loop extrusion to reproduce major features of Hi-C maps is more complicated. We found that one-sided extrusion with uniform association and dissociation of LEFs can generate TADs (*Figure 3b*, right) and 'stripes' (or 'flames,' 'tracks,' or 'lines') (*Fudenberg et al., 2017*; *Fudenberg et al., 2016*; *Vian et al., 2018*) on Hi-C maps (*Figure 3a*). However, one-sided extrusion cannot reliably bring CTCF barriers together, and thus, cannot generate the dots (corner peaks) that are prominent features of Hi-C and micro-C maps (*Krietenstein et al., 2020*) and are reproduced by two-sided extrusion (*Figure 3b*, right and *Figure 3—figure supplement 2*). The presence of unavoidable gaps between LEFs and between LEFs and barriers is the reason for this deficiency. This can be remedied by introducing a comparable number of two-sided LEFs to close gaps (*Figure 3d*, right). One-sided extrusion alone, however, can reproduce dots when undergoing frequent stochastic switches in translocation direction, turning one-sided into effectively two-sided extrusion. Additional mechanisms to generate two-sided or effectively two-sided extrusion have also been proposed (*Davidson et al., 2019*; *Golfier et al., 2020*; *Kim et al., 2020*; *Kim et al., 2019*; *Kong et al., 2020*; *Moevus, 2019*), and gap closure may be achieved by several other mechanisms, as we discuss below in the subsection 'Molecular evidence and plausibility of different modes of SMC function.' Another strategy to eliminate gaps between boundaries and generate dots is to have strongly (>100 fold) biased loading of LEFs at barriers. Loading of cohesin at CTCF sites has been proposed since the two were found to colocalize (*Nichols and Corces, 2015*; *Rubio et al., 2008*). Available experimental evidence, however, argues against loading at CTCF sites; it was previously shown that CTCF is dispensable for cohesin loading (*Parelho et al., 2008*; *Wendt et al., 2008*), and more recently, CTCF-degradation experiments appear to have little effect on the levels of chromatin-associated cohesin (*Busslinger et al., 2017*; *Nora et al., 2019*; *Nora et al., 2017*) and the extent of loop extrusion (*Fudenberg et al., 2017*).

## Bacterial data suggests an 'effectively two-sided' extrusion process

In many bacteria, bSMCs loaded near the origin of replication (by the *parABS* system) generate contacts centered about the *ori-ter* axis, which is visible in Hi-C maps as a secondary diagonal (*Böhm et al., 2020*; *Le et al., 2013*; *Marbouty et al., 2014*; *Umbarger et al., 2011*; *Wang et al., 2017*; *Wang et al., 2015*). The challenge for one-sided loop extrusion models in bacteria is to explain how one-sided (*i.e.*, asymmetric) LEF translocation might generate symmetrically aligned contacts between chromosome arms. Pure one-sided extrusion does not work because it creates a '+"-shape on the contact map instead of a secondary diagonal (*Figure 4c* and [*Miermans and Broedersz, 2018*]). Furthermore, we find that allowing diffusion of the anchor point does not help because this type of asymmetric extrusion cannot promote symmetric juxtaposition of the chromosome arms.

The switching model, however, with a switching time on the order of seconds (<10 s for *B. subtilis* and <20 s for *C. crescentus*, *i.e.*, rates $k_{switch} \geq 0.1$ s$^{-1}$; *Figure 4d*) exhibits the desired effectively two-sided property and naturally creates the desired symmetry of contacts between left and right chromosome arms. Interestingly, if bSMCs function by one-sided extrusion with switching, this constraint suggests that bSMCs can switch their direction of extrusion within a few ATPase cycles (the *B. subtilis* SMC complex has an ATPase rate of 0.7 ATP/s [*Wang et al., 2018*]). Switching, however, has not been observed in single-molecule experiments with yeast condensin SMC complexes, and such fast switching may appear as two-sided extrusion in vitro. We note that it was recently suggested that *B. subtilis* SMCs have two independent motor activities for extrusion (*Brandão et al., 2019*; *Wang et al., 2017*); this observation is consistent with either two-sided extrusion or one-sided extrusion with rapid switching. Thus, our model suggests that microscopically one-sided extrusion can explain juxtaposition of chromosome arms, provided that bSMCs act as effectively two-sided extruders.

## One-sided extrusion may be viable for yeast chromosomes in some, but not all, scenarios

One-sided loop extrusion was first imaged for budding yeast (*S. cerevisiae*) condensins (*Ganji et al., 2018*). Yeast chromosomes are organized differently from chromosomes of higher eukaryotes. In budding yeast, cohesin is responsible for moderate compaction of mitotic chromosomes, while condensin compacts rDNA and proximal regions into insulated domains (*Lazar-Stefanita al al., 2017*; *Schalbetter et al., 2017*) and, in quiescent cells, forms 10–60 kb chromatin domains that silence transcription (*Swygert et al., 2019*). In fission yeast (*S. pombe*), cohesin forms small (<100 kb) domains (*Kim et al., 2016*; *Mizuguchi et al., 2014*; *Tanizawa et al., 2017*), while during mitosis, condensin compacts chromatin by forming larger (100's of kb) domains (*Kakui et al., 2017*; *Kim et al., 2016*; *Tanizawa et al., 2017*).

The ~10 fold linear compaction achievable by pure one-sided loop extrusion is consistent with fluorescence in situ hybridization imaging of yeast mitotic chromosomes (*Guacci et al., 1994*; *Kruitwagen et al., 2018*). Moreover, previous modeling of budding yeast mitotic chromosomes indicated that just ~30–40% coverage by cohesin-extruded loops (*i.e.,*~2 fold linear compaction, *Figure 2—figure supplement 7*) produces chromosome contact maps consistent with those obtained from Hi-C experiments (*Schalbetter et al., 2017*). This lesser degree of compaction generally leads to poorly resolved sister chromatids in our model (*Figure 2b,c (iii), and c (iv)*), but chromatid resolution in yeast could be facilitated by spindle tension (*Lazar-Stefanita et al., 2017*) and the shorter length of yeast chromosomes. These observations could be consistent with compaction by cohesins performing one-sided loop extrusion.

In contrast, one-sided extrusion could account for some, but not all, of the observations of chromatin domains in yeast Hi-C, micro-C, and ChIA-PET experiments. Yeast condensins compact pre- and post-rDNA genomic regions (in *S. cerevisiae*) (*Lazar-Stefanita et al., 2017*; *Schalbetter et al., 2017*) and mitotic chromosomes (*S. pombe*) (*Kakui et al., 2017*; *Tanizawa et al., 2017*) into insulated domains that do not exhibit the dots that are indicative of bringing boundaries together. In a similar manner, fission yeast cohesins organize small chromatin domains without dots (*Kim et al., 2016*; *Mizuguchi et al., 2014*; *Tanizawa et al., 2017*). As shown in *Figure 3b*, pure one-sided loop extrusion can generate domains without dots.

Nonetheless, recent observations of chromatin domains with dots under certain conditions in budding yeast challenge the viability of one-sided extrusion by both condensin and cohesin. In quiescent cells, condensins generate dots at the corners of small (10–60 kb), transcription-silencing domains in micro-C maps (*Swygert et al., 2019*). In exponentially growing cells arrested during mitosis, cohesins can also generate dots in S phase (*Ohno et al., 2019*). This observation suggests that budding yeast condensins and/or cohesins are either effectively two-sided loop extruders or loaded at specific sites because one-sided extrusion alone cannot generate dots (*Figure 3b*). However, a mix of two-sided cohesins and one-sided condensins (*e.g.,* similar to *Figure 3d*, right panels) could generate dots as in micro-C/Hi-C experiments, while remaining consistent with single-molecule experiments.

Cohesin-dependent dots have also been observed at sites of convergent transcription in Hi-C maps when cohesin is overexpressed in G1 (*Dauban et al., 2020*). While such dots can be explained by two-sided extrusion, we also considered the possibility that one-sided extrusion assisted by RNA polymerases that can push one side of an SMC complex (*Lengronne et al., 2004*; *Ocampo-Hafalla and Uhlmann, 2011*). For one-sided extrusion, this effect could in principle generate effectively two-sided (but asymmetric) extrusion, where the slower extruding subunit moves at the speed of transcription (~1 kb/min). For typical cohesin residence times (*Gerlich et al., 2006b*; *Hansen et al., 2017*; *Kueng et al., 2006*; *Tedeschi et al., 2013*; *Wutz et al., 2017*), this model suggests that small loops of 10–60 kb (*Dauban et al., 2020*; *Ohno et al., 2019*) could be generated by the combined activity of loop extrusion and transcription (*Figure 3—figure supplement 8*).

In summary, one-sided extrusion by condensin and cohesin can reproduce some, but not all, of the chromosome organization phenomena observed in yeast. The lower degree of mitotic chromosome compaction (*Guacci et al., 1994*; *Kruitwagen et al., 2018*; *Schalbetter et al., 2017*) and formation of chromatin domains without dots (*Kakui et al., 2017*; *Lazar-Stefanita et al., 2017*; *Mizuguchi et al., 2014*; *Schalbetter et al., 2017*; *Tanizawa et al., 2017*) is consistent with one-sided extrusion by yeast SMC complexes. However, pure one-sided extrusion alone is insufficient to

form dots in Hi-C and micro-C (*Dauban et al., 2020*; *Ohno et al., 2019*; *Swygert et al., 2019*). Consistent with single-molecule experiments, budding yeast condensins could be one-sided, but then cohesins must be two-sided or effectively two-sided in order to generate Hi-C patterns in quiescent cells. In metaphase, budding yeast cohesins may be one-sided extruders, but their interphase activity during exponential growth requires two-sided or effectively two-sided extrusion.

## Molecular evidence and plausibility of different modes of SMC function

Our work identifies two requirements for loop extrusion by SMC complexes to generate known chromosome structures. First, unlooped chromatin gaps between SMC complexes must be closed in order to compact mitotic chromosomes, and they occasionally must be closed between extrusion barriers during interphase to generate enrichment of CTCF-CTCF interactions. Second, particularly in prokaryotes, we find that extrusion must be two-sided or effectively two-sided in order to juxtapose bacterial chromosome arms. Although we studied the switching model in detail, we note that several molecular mechanisms can give rise to such effectively two-sided, gap-closing extrusion. Based on the available experimental evidence, we also considered several physical factors and additional models, discussed below.

### Time and energy requirements for compaction by loop extrusion

Whether loop extrusion can compact and resolve chromosomes within physiological limits is a persistent question for chromosome organization in higher eukaryotes. Previous work on two-sided loop extrusion (*Goloborodko et al., 2016a*) showed that LEFs can compact and resolve metazoan chromosomes (~100 Mb in length) for physiological densities of LEFs (1 per $d$ = 10–30 kb [*Fukui and Uchiyama, 2007*; *Takemoto et al., 2004*; *Walther et al., 2018*]). Compaction and resolution are completed within a few (~5) residence times ($1/k_{unbind}$ ~ 2–10 min [*Gerlich et al., 2006a*; *Terakawa et al., 2017*; *Walther et al., 2018*]), provided that extrusion is fast, *i.e.*, $v$ > 0.2 kb/s (*Goloborodko et al., 2016a*). The extrusion rate of $v \approx 1$ kb/s recently observed in vitro (*Davidson et al., 2019*; *Ganji et al., 2018*; *Golfier et al., 2020*; *Kim et al., 2019*; *Kong et al., 2020*) confirms that loop extrusion is sufficiently rapid to compact metazoan chromosomes during prophase and prometaphase. Moreover, this rate is consistent with expectations from studies of the molecular dynamics of loop-extruding SMC complexes (*Diebold-Durand et al., 2017*; *Marko et al., 2019*).

Furthermore, we can estimate an upper bound on the energy required to compact human chromosomes. Conservatively estimating that condensin or cohesin require two ATP per extrusion step and several attempts to traverse each nucleosome (~150 bp), the ATP cost to extrude 6 Gb is of order 10 x ($6 \times 10^9/150$)~$10^8$ (we assume only ~5 attempts because in vitro extrusion speeds are not measurably altered by nucleosomes [*Kim et al., 2019*; *Kong et al., 2020*]). This upper limit estimate is still less than the ~$10^9$ ATP present in the cell (*Traut, 1994*) and less than the ~$10^9$ ATP/s that the cell produces (*Flamholz et al., 2014*). Moreover, there are only ~$10^5$ cohesins (*Cattoglio et al., 2019*; *Holzmann et al., 2019*) and condensins (*Fukui and Uchiyama, 2007*; *Takemoto et al., 2004*; *Walther et al., 2018*) in each living cell; given an ATPase rate of ~1 s$^{-1}$, we estimate that the rate of actual energy consumption by loop extrusion is ~$10^5$ s$^{-1}$, well within the cell's energy budget. We conclude that genome compaction and organization by loop extrusion is energetically feasible.

### Attractive interactions between LEFs

It has previously been suggested that 3D attractive interactions between LEFs could facilitate compaction of mitotic chromosomes (*Cheng et al., 2015*; *Sakai et al., 2018*). For mitotic chromosomes, our results, along with previous work on polymer combs, suggests otherwise (*Fytas and Theodorakis, 2013*; *Sheiko et al., 2004*). It is possible that SMC complexes may attract each other, but such interactions must be weak enough that the chromosome does not collapse into a spherically symmetric polymer. With weak interactions, however, gaps created by one-sided extrusion cannot be closed, and mitotic chromosomes cannot be formed (*Figure 2—figure supplement 5*). Thus, 3D interactions cannot be the mechanism of chromatin gap closure, and thus, they cannot be essential for mitotic chromosome compaction. For interphase chromosomes, 3D attractions between TAD boundaries (CTCF proteins or their binding sites) could potentially close chromatin gaps. However, 3D attractions would not consistently pair CTCF boundaries in a convergent orientation

(*Fudenberg et al., 2016*; *Sanborn et al., 2015*) nor would they distinguish between proximal and distal TAD boundaries (*Fudenberg et al., 2016*). Furthermore, for both mitotic and interphase chromosomes, attractive 3D interactions would promote trans interactions, contrasting with in vivo observations of condensin-mediated spatial resolution of mitotic chromosomes and cohesin-driven formation of cis loops. All of these points suggest that one-sided loop extrusion together with random cross-bridging of chromatin/DNA segments as in several previous studies (*Bohn and Heermann (2011)*; *Bohn and Heermann (2010)*; *Cheng et al. (2015)*) is not sufficient for compaction and domain formation.

## Regulation of SMC complex residence times

We considered the possibility that interactions between one-sided LEFs and other LEFs or protein factors might alter their residence times, which might facilitate chromosome organization. In simulations of mitotic chromosomes, we found that alterations to LEF residence times due to LEF-LEF interactions do not enhance linear fold compaction (*Figure 2—figure supplement 6*).

## Effects of transcription on loop extrusion

Translocation along DNA by loop-extruding complexes often proceeds in the presence of RNA polymerases that actively translocate as they transcribe genes. We therefore evaluate whether active transcription can help one-sided loop extrusion become effectively two-sided extrusion, or otherwise promote the chromosome organization scenarios studied above. As discussed above, modeling of condensins and RNA polymerases on bacterial chromosomes (*Brandão et al., 2019*), along with experimental evidence for other cell types (*Busslinger et al., 2017*; *Dauban et al., 2020*; *Davidson et al., 2016*; *Glynn et al., 2004*; *Heinz et al., 2018*; *Lengronne et al., 2004*), suggests that translocating RNA polymerases can push translocating SMC complexes, and thus alter chromosome organization. While transcription can occur during mitosis, inhibiting transcription does not visibly alter mitotic chromosome compaction (*Palozola et al., 2017*). Furthermore, only condensin and a few other protein factors are required to form mitotic chromosomes in vitro (*Shintomi et al., 2017*; *Shintomi et al., 2015*). Therefore, pushing of condensins by RNA polymerases cannot be the primary mechanism underlying the predicted requirement for effectively two-sided loop extrusion in mitosis. In contrast, formation of cohesin-dependent dots between convergent genes in budding yeast Hi-C (*Dauban et al., 2020*) requires either effectively two-sided extrusion by cohesin or a hypothetical mechanism in which one-sided extrusion is assisted by transcription: one-sided cohesins could become effectively two-sided if RNA polymerase (translocating at $v \sim 1$ kb/min) is able to efficiently push the passive side of the cohesin complex (*Figure 3—figure supplement 8*). This assistance would further require specific orientations of multiple genes (*Figure 3—figure supplement 8*). In bacteria (*B. subtilis* and *C. crescentus*), the ability of bSMCs to juxtapose chromosome arms is largely unaffected by transcription inhibition (*Brandão et al., 2019*; *Tran et al., 2017*; *Wang et al., 2017*). Additionally, pushing of bSMCs by RNA polymerases cannot drive chromosome arm juxtaposition because genes are not universally transcribed from *ori* to *ter*; as such, RNA polymerase together with one-sided extrusion would be unable to juxtapose the entire length of two chromosomal arms. Furthermore, condensin seems to be able to traverse highly transcribed genes within mere seconds (*Brandão et al., 2019*). Thus, for bacteria, transcription is also not an essential driver of effectively two-sided loop extrusion. Altogether, transcription cannot be the driving force of metazoan mitotic chromosome compaction and bacterial chromosomal arm juxtaposition, but it could help drive effectively two-sided, but asymmetric, extrusion by cohesins in yeast in some specific scenarios.

## Diffusive slip links are not consistent with the experimental data

It has previously been proposed that SMC complexes with purely diffusive subunits might organize interphase TADs (*Brackley et al., 2017*; *Yamamoto and Schiessel, 2017*). In this model, cohesins with two diffusive subunits are loaded at a loading site. Osmotic pressure arising from the successive loading of multiple cohesins at the loading site biases loop growth such that boundary elements (*i. e.*, CTCFs) may be brought together. However, targeted loading of LEFs in vertebrate cells has not been observed. Moreover, our modeling shows that even a semi-diffusive model fails to compact and resolve mitotic chromosomes (*Figure 2d*), generate TADs with dots (*Figure 3d*, left), or

juxtapose bacterial chromosome arms (*Figure 4c*). Consistently, previous modeling demonstrated that slip links could only juxtapose bacterial chromosome arms at unphysiologically high densities (*Miermans and Broedersz, 2018*). Thus, diffusive slip links are not sufficient to account for various chromosome organization phenomena.

## Oligomerization of SMC complexes

SMC complex oligomerization could facilitate chromosome organization by suppressing gap formation and/or promoting symmetric extrusion in various scenarios. In eukaryotes, in situ amino acid crosslinking (*Barysz et al., 2015*) and in vitro gel filtration (*Keenholtz et al., 2017*) suggest that condensins can oligomerize. Several experiments similarly suggest that cohesin may form oligomeric complexes in vitro (*Kim et al., 2019*) or in vivo (*Cattoglio et al., 2019*; *Eng et al., 2015*; *Nagy et al., 2016*; *Zhang et al., 2008*). Formation of such complexes could lead to effectively two-sided extrusion and gapless chromosome compaction. In prokaryotes, such as *E. coli* (which have MukBEF complexes, SMC complex homologs), experiments show that MukBEF forms dimers of complexes (*Badrinarayanan et al., 2012*) linked by the kleisin molecule, MukF (*Zawadzka et al., 2018*). MukBEF complexes promote long-ranged contacts within *E. coli* chromosome arms (*Lioy et al., 2018*), and they are proposed to function by two-sided loop extrusion. Dimerization has also been suggested for other bacterial SMC complexes (*Brandão et al., 2019*; *Diebold-Durand et al., 2017*; *Tran et al., 2017*; *Wang et al., 2018*), but it is still unknown whether bSMCs in well studied organisms like *C. crescentus* and *B. subtilis* dimerize in vivo. Functional dimerization of bSMCs in vivo could be directly tested by photobleaching experiments with endogenous fluorescently tagged versions of bSMC, as in *Badrinarayanan et al. (2012)*. Additionally, to determine whether MukBEF dimerization is needed for DNA loop formation, we suggest a Hi-C experiment on a MukBEF mutant deficient in dimerization. If long-ranged chromosome interactions and proliferation under fast-growth conditions persist, then dimerization is not required for MukBEF function. These experiments could therefore investigate the possible functional role of SMC complex oligomerization in loop extrusion.

## Two-sided extrusion and LEF traversal

Recent single-molecule experiments have reported the first observations of two-sided and effectively two-sided loop extrusion. It has been shown that ~ 80% of human condensin I and ~50% of human condensin II complexes perform two-sided DNA loop extrusion in vitro (*Kong et al., 2020*). This finding suggests that human condensins in vivo might satisfy constraints predicted by previous theory (*Banigan and Mirny, 2019*) and new simulations (*Figure 2—figure supplement 1*), which show that ~ 85% of LEFs must be two-sided in order to achieve 1000-fold linear chromatin compaction and robust 3D compaction of mitotic chromosomes. Similarly, recent single-molecule experiments observe mostly two-sided extrusion by human and *Xenopus* cohesin (*Davidson et al., 2019*; *Golfier et al., 2020*; *Kim et al., 2019*), which is consistent with our finding that > 50% two-sided extrusion is needed to reproduce the 'dots' that reflect elevated CTCF-CTCF contact frequency in interphase (*Figure 3d*, right).

Other single-molecule experiments have shown that yeast condensins can traverse each other, which in turn may act as effectively two-sided extruders (*Kim et al., 2020*). We simulated and analyzed a simple realization of this scenario, in which condensins, cohesins, or bSMCs can pass each other as they translocate along DNA/chromatin. For simulations of mitosis, this leads to loop coverage that increases exponentially with $\lambda/d$ and compacted rod-like chromosomes (*Figure 2f–g*). However, our model with LEFs traversing each other generates many pseudoknots, and thus, linear spatial ordering of the mitotic chromosome is not maintained on length scales comparable to the loop size, $\ell \approx \lambda$, which may be >100 kb (as estimated from measured condensin speed [*Ganji et al., 2018*; *Kim et al., 2020*] and turnover rate [*Gerlich et al., 2006a*; *Terakawa et al., 2017*; *Walther et al., 2018*]). For interphase simulations, the ability of cohesins to pass each other increases the strength of dots (corner peaks) as compared to pure one-sided extrusion. However, dots are not as strong as they are with two-sided extrusion, and they only appear for a high cohesin densities and/or processivities (*Figure 3—figure supplement 7*). In contrast, LEF traversal does not facilitate juxtaposition of bacterial chromosome arms because the one-sided LEFs maintain contacts between the origin and distal regions of the chromosome (*Figure 4—figure supplement 4*).

Moreover, several questions remain about the in vivo relevance of LEF traversal, and the formation of the 'Z-loop' structure. We assumed that each LEF may traverse any other LEF that it encounters, but it is unknown how SMCs contributing to Z-loop structures actually interact. A more restrictive set of traversal rules could severely limit linear compaction and corner peak formation. For example, if each active subunit can only traverse a single anchored subunit, then linear compaction is limited to 50-fold (following arguments for the 'weak pushing' model, see Appendix 1 and *Figure 2—figure supplement 3*). In addition, it is unknown how Z-loop formation is altered when condensins or cohesins extrude chromatin instead of DNA. Thus, while our preliminary modeling suggests that effectively two-sided extrusion by Z-loops might compact mitotic chromosomes and pair CTCF sites, a number of experimental and theoretical factors remain unexplored.

## Predictions and suggestions for future experiments

In *Table 1*, we list possible mechanisms of loop extrusion and whether they are able to reproduce in vivo experimental observations; however, many of these mechanisms have not yet been observed or tested. Single-molecule experiments (*Davidson et al., 2019*; *Ganji et al., 2018*; *Golfier et al., 2020*; *Kim et al., 2020*; *Kim et al., 2019*; *Kong et al., 2020*) could assay different types of SMC complexes from a range of organisms in order to establish which loop extrusion models are applicable. We predict that SMC complexes in vivo may constitute effectively two-sided motors or exhibit biased loading in order to robustly organize and compact chromatin. However, a variety of microscopic (molecule-level) modes of extrusion may achieve the same macroscopic organization of the chromosomal DNA.

We make several testable predictions. First, if switching of extrusion direction is observed, switching should be fast (occurring at least once per 10 s for bSMCs and at least once per minute for human SMC complexes cohesins and condensins). In addition, we predict that if a mixture of one-sided and two-sided extrusion is observed for a population of SMC complexes, then the fraction of two-sided extrusion should be at least 50% for cohesin and at least 80% for condensin (*Table 1*). We also predict that bSMCs from eubacteria are either two-sided monomeric complexes or a dimer of complexes that translocate in opposing directions, enlarging a loop and resulting in two-sided extrusion.

A few other types of experiments are critical to perform at the single-molecule level in vitro; these would be difficult to test in vivo by microscopic and biochemical methods. We suggest: 1) testing how SMC complexes interact with one another when they meet on the same chromatin/DNA substrate in vivo, as we show that LEF traversal can lead to effective compaction; 2) testing whether/what fraction of SMC complexes do one-sided or two-sided extrusion under different conditions, such as at various salt concentrations and/or with molecular crowding agents; and 3) testing whether specific factors, such as chromatin conformations (*e.g.*, supercoils or Holliday junctions) or proteins (*e.g.*, other SMC complexes or CTCF), affect mechanisms of extrusion.

Finally, we note that there may be differences in functionality among condensins of different species or physiological scenarios. For example, it has been hypothesized that yeast condensins could be one-sided because they do not need to linearly compact mitotic chromosomes 1000-fold (*Banigan and Mirny, 2019*). If yeast condensin is fundamentally different from human condensin in function, its use in cell-free chromosome assembly systems (*Shintomi et al., 2017*; *Shintomi et al., 2015*) should result in long, poorly folded chromosomes relative to those with condensin II only. Similarly, mutations that bias condensin activity towards one-sided extrusion could lead to catastrophic under-compaction of human chromosomes, failure to decatenate chromosomes (*Martin et al., 2016*), DNA damage, aneuploidy, developmental disorders (*Martin et al., 2016*), and cancer (*Mazumdar et al., 2015*; *Woodward et al., 2016*).

## Conclusion

The loop extrusion model has been hypothesized to explain a variety of chromosome organization phenomena, but until recently had remained a hypothesis. Experimental work on yeast condensins (*Ganji et al., 2018*; *Kim et al., 2020*) has observed that loop extrusion by yeast condensins occurs in a one-sided manner. Theory and simulations of one-sided loop extrusion (*Banigan and Mirny, 2019*; *Miermans and Broedersz, 2018*) challenge the generality of this observation. We have shown that pure one-sided loop extrusion generally is unable to reproduce a variety of chromosome

organization phenomena in different organisms and scenarios. Instead, loop extrusion should be 'effectively two-sided' and/or have the ability to robustly eliminate unlooped chromatin gaps to organize chromosomes; in accord with this, recent experimental data indicate that human condensins and human and *Xenopus* cohesins are capable of acting in a two-sided manner (*Davidson et al., 2019*; *Golfier et al., 2020*; *Kim et al., 2019*; *Kong et al., 2020*). Additionally, among the models we explored, the switching model is an example that meets these requirements. Nonetheless, experimental evidence suggests that different organisms are likely to achieve macroscopic chromosome organization through diverse microscopic mechanisms. While loop extrusion remains a unifying model for chromosome organization across different domains of life, various to-be-determined microscopic mechanisms could underlie these phenomena.

## Materials and methods

### Basic model

Stochastic simulations of loop-extrusion dynamics are performed with $N$ LEFs on a lattice of length $L$. There are several types of events. LEFs bind to the chromatin lattice at rate $k_{bind}$ by occupying two adjacent lattice sites and LEFs unbind at rate $k_{unbind}$. When an active subunit of a LEF makes a step, it occupies the site that was immediately adjacent to it, which frees the lattice site that it previously occupied. Directional stepping by an active subunit occurs at speed $v$ and proceeds in the direction away from the other LEF subunit. Diffusive stepping occurs in either direction at loop-size-dependent rate $v^{\pm}_{diff}(\ell)$. When a one-sided LEF switches its active extrusion direction, the active subunit becomes passive and vice versa. Switches occur at a rate $k_{switch}$. In interphase simulations, LEF subunits may stall upon encountering a correctly oriented CTCF site. This occurs with probability $p_{stall}$. Each simulation consists of a chromatin polymer with $L$ sites and a fixed number, $N_b$, of LEFs that populate the sites at low density, $N_b/L \leq 0.05$. The simulation code is publicly available at https://github.com/mirnylab/one_sided_extrusion (*Banigan et al., 2020*; copy archived at https://github.com/elifesciences-publications/one_sided_extrusion).

### Event-driven (Gillespie) simulations for linear compaction

1D stochastic simulations of loop-extrusion dynamics modeling mitotic chromosome compaction for pure one-sided, two-sided, switching, and pushing models are performed with $N$ LEFs on a lattice of length $L$, with $L$ = 60000 sites and $100 < N < 3000$. Each site is taken to be $a$ = 0.5 kb.

We use the Gillespie algorithm to determine the time that each kinetic event – binding, unbinding, directional stepping, and switching – occurs (*Gillespie, 1977*; *Goloborodko et al., 2016b*). Events are executed in temporal order, and after an event occurs, we compute the lifetimes of new events that become permissible (*e.g.*, a LEF step that becomes possible because another LEF has moved). Simulations are run for $t_{sim}$ = 400 max$((1/k_{unbind}+1/k_{bind})$, $L/v+1/k_{bind})$, and data is recorded for the second half of the simulation, long after the onset of the steady-state, for at least three simulations per parameter combination.

### Fixed-time-step simulations for LEF dynamics

For 1D simulations of chromosome compaction in the semi-diffusive model, 1D simulations of compaction with LEF traversal, 3D polymer simulations of chromosome compaction with all models, interphase TAD formation, and 1D simulations of LEF dynamics on bacterial chromosomes, we use a fixed-time-step Monte Carlo algorithm instead of the Gillespie algorithm. This algorithm facilitates coupling of LEF kinetics to the loop architecture (for the semi-diffusive model) and/or 3D polymer conformation (for polymer simulations). Here, each event is modeled as a Poisson process; at each LEF time step $dt$, an event is executed with probability $k_i dt$, where $k_i$ is the rate of event $i$. In the semi-diffusive model, the passive diffusive stepping rate for a LEF is $v^{\pm}_{diff}(\ell)=v_{diff}\ e^{\mp(3/2)\ (a\ /\ \ell)}$, which is updated when the size of either the loop associated with the LEF or any loop in which the LEF is nested changes in size. The expression for $v^{\pm}_{diff}(\ell)$ is a discretization of $v^{\pm}_{diff}(\ell)=v_{diff}\ e^{\mp f\ a\ /\ kT}$. Here, $f$ = -d$U$/d$\ell$ = (3/2) $kT$ ln($\ell/a$) defines the entropic force arising from loop configurational entropy (*e.g.*, see *Brackley et al. (2017)*.

## Simulations of mitotic chromosomes

For fixed-time-step simulations of mitotic chromosomes, $L$ = 30000, $N$ = 750, and $a$ = 0.5 kb, which is assumed to be 30 nm in diameter (~3 nucleosomes). At least three simulations per parameter combinations are run for >40 residence times, and linear compaction is measured after 20 residence times. Probe radius $r_{hull}$ = 600 nm was used to calculate concave hulls.

## Simulations of interphase chromosomes

For simulations of interphase, we simulate a chain with three different TAD sizes of 100, 200, and 400 monomers. This system of 700 monomers in total is repeated 6 or eight times, giving a total size of 4200 monomers (for computing dot strengths) or 5600 monomers (for computing contact maps and scalings). When LEFs encounter a CTCF site, they are stalled (i.e. they stop moving until they are unloaded), with a probability of 80% (*Fudenberg et al., 2016*). From the scalings, we determined that one monomer corresponds to 2 kb (*Figure 3—figure supplement 1*).

We used a total of 4000 conformations to compute contact maps, scalings or dot strengths. For computing the contact maps, we used a contact radius of 5 monomers. Dot strengths are computed as follows: first, we compute observed-over-expected of a contact map (we divide out the distance dependence, by dividing each diagonal by its average [*Lieberman-Aiden et al., 2009*]), then we compute the strength of a dot of a particular TAD (*Figure 3—figure supplement 2*) and last, we compute the average of all the dots (each of which appears six times on one map).

In contrast to mitotic compaction, $\lambda$ and $d$ are varied separately for interphase chromosomes, because the dot strengths depend on $\lambda$ and $d$ separately, as well as the distance between two CTCF sites, $d_{CTCF}$. Based on contact probability scalings (*Figure 3—figure supplement 1*) and experimental observations, we consider a separation between loop extruders of $d$ = 200 kb and a processivity of $\lambda$ = 200 kb (*Cattoglio et al., 2019*; *Fudenberg et al., 2016*; *Holzmann et al., 2019*) in the main text, and we consider other parameter values in the figure supplements. Furthermore, we choose typical TAD sizes of 200 and 400 kb (*Rao et al., 2014*). For simulations of Wapl depletion conditions, we use $d$ = 200 kb and $\lambda$ = 2 Mb (*Gassler et al., 2017*; *Nuebler et al., 2018*).

## Simulations of bacterial chromosomes

We simulate loop extrusion on bacterial chromosomes using the fixed-time-step simulations for LEF dynamics described above. LEFs are allowed to randomly load on a lattice of $L$ = 4000 sites, where each lattice site corresponds to ~1 kb of DNA. LEFs have a strong bias to bind one site at the center of the lattice to mimic the effect of a single *parS* site near the origin of replication in bacterial chromosomes. The relative probability of loading at the simulated *parS* site was ~40,000 times stronger than that of every other site, *i.e.*, if the relative probability of loading at the simulated *parS* is 1, then the total relative probability to load on *any* other site is 0.1 $L$. As a result, the overall preference to bind the *parS* site over all other genomic loci is approximately 10-fold.

Bacterial LEFs were simulated as deterministic extruders with a stochastic dissociation rate $k_{unbind}$ = 2/$L$ to approximate the steady decrease in bSMC density away from the *ori* observed via ChIP-seq (*i.e.*, bSMC density at the *ter* region is ~1/3 of the value at *ori*) (*Wang et al., 2017*). In addition to a stochastic (position-independent) dissociation rate, LEFs automatically unbind if one of the subunits reached the edge of the lattice, *i.e.*, the *ter* region; *ter* was set to lattice positions 0–3 and 3996–3999 (*i.e.*, diametrically opposite to the *parS* site at lattice site 2000).

## Polymer simulations with OpenMM

To model the 3D dynamics of polymers loaded with LEFs, we performed polymer molecular dynamics simulations in OpenMM (*Eastman et al., 2017*; *Eastman et al., 2013*; *Eastman and Pande, 2010*) using a custom, publicly available library, openmm-polymer (available at https://github.com/mirnylab/openmm-polymer-legacy; (*Imakaev et al., 2020*), coupled with the fixed-time-step LEF simulations described above and in *Fudenberg et al. (2016)*; *Goloborodko et al. (2016a)*.

In the polymer simulation, a LEF crosslinks the sites that it occupies together. LEF positions are evolved as described above. After each time step of LEF dynamics, the polymer simulation is evolved via Langevin dynamics for 200 or 250 time steps (for interphase and mitosis, respectively) with $dt$ = 80.

Polymers are constructed of $L$ consecutive subunits bonded via the pairwise potential:

$$U_b(r) = \frac{k}{2}(r-b)^2$$

where $r = r_i\ r_j$ is the displacement between monomers $i$ and $j$, $k = 2\ kT\ /\ \delta^2$ is the spring constant, $\delta = 0.1$, and $b$ is the diameter of a monomer. For mitotic chromosome simulations, $b = 30$ nm; for other scenarios, it is unnecessary to assign a value to $b$. Monomers crosslinked by a LEF are held together by the same potential. Weakly repulsive excluded volume interactions between monomers are modeled as:

$$U_{exc}(r) = \frac{\varepsilon_{exc}}{\varepsilon_m}\left(\frac{r}{\sigma}r_m\right)^{12}\left(\left(\frac{r}{\sigma}r_m\right)^2 - 1\right) + \varepsilon_{exc},$$

for $r<\sigma$ with $\sigma=1.05b$, $r_m = \sqrt{6/7}$, $\varepsilon_m$=46656/823543, and $\varepsilon_{exc}$=1.5 $kT$. For simulations of mitotic chromosomes with 3D attractive interactions, monomers interact through the potential:

$$U_{att}(r) = -\frac{\varepsilon}{\varepsilon_m}\left(\frac{r}{\sigma}r_m\right)^{12}\left(\left(\frac{r}{\sigma}r_m\right)^2 - 1\right) + \varepsilon,$$

for $\sigma < r < 2b$ and $\varepsilon$ is a parameter to be varied.

At the beginning of each simulation, the polymer is initialized as a random walk and monomers are initialized with normally distributed velocities, so that the temperature is $T$. The system is thermostatted by intermittent rescaling of velocities to maintain temperature $T$.

### Contact probability calculations in the Gaussian chain approximation

To compute contact maps for bacterial chromosomes, the contact frequency was calculated from the equilibrium contact probability for a Gaussian chain. This theoretical model agrees well with polymer molecular dynamics simulations (Appendix 3 and *Figure 4—figure supplements 6, 7*). Briefly, contact probability between two sites on a Gaussian chain scales with $s^{-3/2}$, where $s$ is the linear distance between the sites, excluding any loops between the two sites. Sites within the same loop obey this scaling relation with an effective $s$, $s_{eff}$, substituted for $s$ in the scaling relation; $s_{eff} = s(1\ s/\ell)$, where $\ell$ is the loop size. For sites in different loops, $s$ in the scaling relation is replaced by the sum of the effective lengths of the regions connecting the two sites (see Appendix 3 for details). These relative contact probabilities are used to compute the contact maps for bacterial chromosome simulations. Contact maps are generated using contacts from 50,000 to 100,000 different LEF conformations.

## Acknowledgements

We thank current and former members of the Mirny group for ongoing discussions and critical feedback; we particularly thank Anton Goloborodko and Maxim Imakaev for helpful discussions and sharing code and Aleksandra Galitsyna for discussions. We also thank Sergey Belan, Xavier Darzacq, Cees Dekker, Job Dekker, and Christian Haering for helpful discussions. This work was supported by the NIH Center for 3D Structure and Physics of the Genome of the 4DN Consortium (U54DK107980) and the Physical Sciences-Oncology Center (U54CA193419). EJB, AAB, HBB, and LAM were also supported by the NIH through GM114190.

## Additional information

### Funding

| Funder | Grant reference number | Author |
| --- | --- | --- |
| National Institutes of Health | U54DK107980 | Edward J Banigan<br>Aafke A van den Berg<br>Hugo B Brandão<br>John F Marko<br>Leonid A Mirny |

| National Institutes of Health | U54CA193419 | Edward J Banigan<br>Aafke A van den Berg<br>Hugo B Brandão<br>John F Marko<br>Leonid A Mirny |
| National Institutes of Health | GM114190 | Edward J Banigan<br>Aafke A van den Berg<br>Hugo B Brandão<br>Leonid A Mirny |

The funders had no role in study design, data collection and interpretation, or the decision to submit the work for publication.

## Author contributions

Edward J Banigan, Aafke A van den Berg, Hugo B Brandão, Conceptualization, Software, Formal analysis, Investigation, Interpretation of results; John F Marko, Conceptualization; Leonid A Mirny, Conceptualization, Supervision, Funding acquisition, Interpretation of results

## Author ORCIDs

Edward J Banigan  https://orcid.org/0000-0001-5478-7425
Aafke A van den Berg  https://orcid.org/0000-0003-2991-9478
Hugo B Brandão  https://orcid.org/0000-0001-5496-0638
Leonid A Mirny  https://orcid.org/0000-0002-0785-5410

## Decision letter and Author response

Decision letter https://doi.org/10.7554/eLife.53558.sa1
Author response https://doi.org/10.7554/eLife.53558.sa2

# Additional files

## Supplementary files

• Transparent reporting form

## Data availability

Simulation and analysis code used to produce and analyze data has been made publicly available on GitHub. Methods and code documentation explains usage. In Figure 3, we show Hi-C data from another publication with GEO accession number GSE96107. In Figure 3 - figure supplement 8, we show Hi-C data from another publication with Bioproject accession number PRJNA427106. In Figure 4, we show Hi-C data from another publication with GEO accession number GSE68418.

The following previously published datasets were used:

| Author(s) | Year | Dataset title | Dataset URL | Database and Identifier |
|---|---|---|---|---|
| Bonev B, Mendelson Cohen N, Szabo Q, Fritsch L, Papadopoulos G, Lubling Y, Xu X, Lv X, Hugnot J, Tanay A, Cavalli G | 2017 | Multi-scale 3D genome rewiring during mouse neural development | https://www.ncbi.nlm.nih.gov/geo/query/acc.cgi?acc=GSE96107 | NCBI Gene Expression Omnibus, GSE96107 |
| Ohno M | 2017 | Studies on chromosome structure at sub-nucleosome level | https://www.ncbi.nlm.nih.gov/bioproject/?term=PRJNA427106 | NCBI BioProject, PRJNA427106 |
| Wang X, Rudner DZ | 2015 | Condensin promotes the juxtaposition of DNA flanking its loading site in Bacillus subtilis | https://www.ncbi.nlm.nih.gov/geo/query/acc.cgi?acc=GSE68418 | NCBI Gene Expression Omnibus, GSE68418 |

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

## Appendix 1

# LEF pushing models

## Descriptions of the 'strong' and 'weak' pushing models

We consider two variations of 'pushing' models, in which passive subunits of a loop-extruding factor (LEF) may be pushed by the active subunit of another LEF. As in the other one-sided extrusion models, LEFs are comprised of one active subunit and one passive subunit. When an active subunit of the first LEF encounters a passive subunit, the active subunit may continue translocation by forcing the passive subunit off of its chromatin polymer lattice site and onto the adjacent site, in the direction of active translocation. In the 'weak' pushing model, an active subunit can push a single passive subunit onto adjacent unoccupied sites (**Figure 2— figure supplement 4a**, top). In the 'strong' pushing model, if multiple passive subunits are adjacent to each other, an active subunit behind the consecutive chain of adjacent passive subunits may directionally push the passive subunits, provided that there is an unoccupied site at the other end of the chain (**Figure 2—figure supplement 4a**, top).

## Mean-field theoretical calculation for the weak pushing model

Using the mean-field theory previously developed for loop extrusion in the limit of large $\lambda/d$ (**Banigan and Mirny, 2019**), we can calculate the maximum attainable linear fold compaction in the weak pushing model (there is no compaction limit for the strong pushing model, because all gaps can be closed for sufficiently large $\lambda/d$). Specifically, this calculation assumes that the processivity, $\lambda$, is large ($\lambda \gg d$) and the system is in steady state. To determine the fraction, $f$, of chromatin that is compacted into loops, we must determine the frequency of gaps, which remain if adjacent LEFs are divergently oriented (*i.e.*, $\leftarrow\rightarrow$). As described below, we may then compute the equivalent fraction of LEFs that are effectively two-sided, and thus, the associated maximum attainable linear fold compaction.

### Review of mean-field theory for loop extrusion

In the pure one-sided model, there is one gap for every four loops, which leads to the equation:

$$N_p \ell + \frac{N_p}{4} g = L, \tag{1}$$

where $N_p$ is the number of parent LEFs (*i.e.*, LEFs found at the bases of chromatin loops), $\ell$ is the mean length of a loop, $g = d$ is the mean gap size, and $L$ is the length of the chromatin polymer.

Two additional equations will be needed to solve the weak pushing model. From **Equation 1**, we can write:

$$f = \frac{4\ell}{4\ell + g} = 1 - \frac{N_p}{4(N_p + N_c)}, \tag{2}$$

where $N_c$ is the total number of nested child LEFs. In addition, by solving the equations for the steady-state binding/unbinding kinetics of LEFs, we find:

$$N_c = \frac{f - \alpha}{1 - f} N_p, \tag{3}$$

where $\alpha$ is the fraction of parent LEFs that have a child LEF nested within.

From these equations, as described in **Banigan and Mirny, 2019** we find $f = (3 + 4\ln 4)/(4 + 4\ln 4) = 0.895$. Since linear fold compaction is defined as:

$$\mathcal{FC} = \frac{1}{1-f},\tag{4}$$

we have $FC \approx 10$.

The theory can be extended to compute linear compaction for systems that include two-sided or effectively two-sided LEFs. If a fraction, $\phi$, of LEFs are (effectively) two-sided, *Equation 1*, relating loops, gaps, and polymer length becomes:

$$N_p \ell + \frac{N_p(1-\phi)^2}{4} g = L.\tag{5}$$

The maximum fraction compacted is then given by:

$$f = \frac{3 + 2\phi - \phi^2 + 4\ln(4(1-\phi)^{-2})}{4 + 4\ln(4(1-\phi)^{-2})}.\tag{6}$$

## Application of mean-field theory to the weak pushing model

In the weak pushing model, some gaps left by one-sided extrusion may be closed if at least one of the two 'parent' LEFs adjacent to the gap has a nested 'child' LEF that is oriented so that its active subunit translocates toward the passive subunit of the parent LEF. To compute the fraction compacted, $f$, we modify *Equation 1* to properly describe the frequency of unlooped gaps along the chromosome because some gaps may be closed by nested child LEFs.

We begin by computing the probability that a particular gap will be closed by a nested LEF. Because we consider a 'weak' pushing model in which an active subunit may only push a single passive subunit (*Figure 2—figure supplement 4a*, top), we only need to consider the top level of LEF nesting. Each parent LEF has a probability $\alpha$ of having a nested child LEF. The child LEF has a 50% chance of being oriented so that it actively extrudes toward the passive subunit of the parent LEF. This configuration closes unlooped gaps. Thus, each LEF in a *potentially* gapped configuration does not close the gap with probability $1 - \alpha/2$. Since each potential gap is bordered by two parent LEFs, we have the following equation for gaps and loops:

$$N_p \ell + \frac{N_p}{4}(1-\alpha/2)^2 g = L.\tag{7}$$

Paralleling the analysis in *Banigan and Mirny, 2019*, we can rewrite this equation as:

$$f = 1 - \frac{N_p}{4(1-\alpha/2)^{-2}(N_p + N_c)},\tag{8}$$

and use *Equation 3* to find $\alpha = 2(2\sqrt{3} - 3) = 0.928$. By substituting into *Equation 7* and comparing to *Equation 5*, we find that weak pushing corresponds to an effective two-sided fraction of $\phi = 2\sqrt{3} - 3 = 0.464$. This leaves an average of $n_g/n_\ell = (1/4)(1 - \alpha/2)^2 = 0.072$ gaps per loop (*Figure 2—figure supplement 4c*, brown dashed line). Substituting into *Equation 6*, we find:

$$f = \frac{\mathrm{arccosh}7 + 4\sqrt{3} - 6}{1 + \mathrm{arccosh}7} = 0.980,\tag{9}$$

which corresponds to $\mathcal{FC} = 51$-fold linear compaction (*Figure 2—figure supplement 4b*, brown dashed line).

## Appendix 2

### Linear compaction by LEFs that can traverse each other

In the main text, we considered a model in which LEFs may traverse each other, that is, they do not act as barriers to each other. This is one possible many-LEF theoretical model for the Z-loops observed in **Kim et al., 2019**. We may compute linear compaction, $\mathcal{FC}$, as defined in **Equation 4**, by computing the fraction of chromatin that is extruded into loops. Since LEFs are essentially invisible to each other in this model, we may compute loop coverage by randomly placing loops of size $\lambda$ (the processivity) on a polymer of length $L$. We will first compute the fraction of the polymer that is not extruded into loops and then subtract this result from 1.

First consider a randomly chosen loop on the polymer and a random infinitesimal region of length $du$. The probability that this infinitesimal region is not covered by the particular loop $p = (L-\lambda)/L$. Since LEF (and thus, loop) positions are independent of each other in this model, the probability that the region $du$ is not covered by *any* of the $N$ loops is $p^N$. Integrating over the entire polymer, we find the total average uncovered length:

$$\langle u \rangle = \int_0^L du \left(\frac{L-\lambda}{L}\right)^N = L\left(\frac{L-\lambda}{L}\right)^N. \tag{10}$$

Therefore, the fraction extruded into loops is:

$$f = 1 - \frac{\langle u \rangle}{L} = 1 - \left(\frac{L-\lambda}{L}\right)^N = 1 - e^{-\lambda N/L}. \tag{11}$$

Using **Equation 4** and noting that $d = L/N$, fold compaction grows exponentially with $\lambda/d$:

$$\mathcal{FC} = (1-f)^{-1} = e^{\lambda/d}. \tag{12}$$

1000-fold linear compaction in this model is achieved for $\lambda/d = 6.9$.

## Appendix 3

# Generating Hi-C-like contact maps analytically

We devised a method of quickly generating Hi-C-like contact maps assuming the polymer is an equilibrium Gaussian chain. Contact maps can be rapidly generated from a list of SMC complex positions. This analytical method allows us to generate Hi-C-like maps quickly, circumventing the need to perform a more computationally intensive 3D Brownian or molecular dynamics (MD) polymer simulation. In *Figure 4—figure supplement 8*, we provide an overview of the method for calculating contact probability between two genome loci. We treat the cases in which SMC complexes do not form pseudoknots and SMC-mediated physical contacts between two monomers of the polymer chain have a root-mean-squared distance similar to the monomer length. To compute Hi-C-like contact maps, we compute the effective genomic distance between any two points on the chain. The effective distance is the harmonic mean of the two shortest paths that can be taken between the two points within a looped segment (see *Figure 4—figure supplement 8*). We present our findings in the context of generating bacterial Hi-C maps, and we validate the method by direct comparison to an MD simulation of a 3D polymer.

## Contact probability of a linear chain

A Gaussian chain in one dimension with $N$ segments of mean square length $b^2$, has a configurational probability density given by:

$$P(r_1, ..., r_N) = A \exp\left(\frac{r_1^2}{2b^2}\right) \exp\left(\frac{-|r_2 - r_1|^2}{2b^2}\right) ... \exp\left(\frac{-|r_N - r_{N-1}|^2}{2b^2}\right)$$

$$= A \prod_{i=1}^{N} g(r_i - r_{i-1}),$$

(13)

where $g$ is defined to be the Gaussian function, and $r_0$ is set to the origin:

$$g(r_i - r_{i-1})r_0 = \exp\left(\frac{-|r_i - r_{i-1}|^2}{2b^2}\right); \ r_0 = 0.$$

(14)

The normalization factor $A$ can be calculated by integrating over all $r_i$ by making a change of variables:

$$A^{-1} = \int_{-\infty}^{\infty} dx_1 ... \int_{-\infty}^{\infty} dx_N \prod_{i=1}^{N} g(x_i),$$

(15)

$$x_i = r_i - r_{i-1} \ \forall i \in [1, N].$$

(16)

The Jacobian of this transformation is unity, since this is an upper triangular matrix of ones on the diagonal. Thus, we get:

$$A^{-1} = \prod_{i=1}^{N} \int_{-\infty}^{\infty} dx_i \exp\left(\frac{-x_i^2}{2b^2}\right) = \left(2\pi b^2\right)^{N/2}$$

(17)

by using the identity:

$$\int_{-\infty}^{\infty} \exp(-ax^2) dx = \sqrt{\frac{\pi}{a}}.$$

(18)

To calculate the cyclization probability of the linear chain of $N$ segments, we first calculate $P(r_N)$ and set $r_N = 0$ and integrate over the distribution of all 'internal' steps $\{r_1, ..., r_{N-1}\}$. This

calculation is more easily solved using the convolution theorem and Fourier transform pairs defined by the convention below:

$$
\begin{aligned}
\int_{-\infty}^{\infty} g(x)g(t-x)dx &= \mathcal{F}^{-1}[\mathcal{F}(g(x))\mathcal{F}(g(t-x))] \\
&= \mathcal{F}^{-1}[\tilde{G}(k)\cdot\tilde{G}(k)]
\end{aligned}
\tag{19}
$$

where the Fourier transforms are defined by:

$$
\begin{aligned}
\mathcal{F}[g(x)] &= \tilde{G}(k) = \int_{-\infty}^{\infty} g(x)\exp(-ik\cdot x)dx \\
\mathcal{F}^{-1}[\tilde{G}(k)] &= \frac{1}{2\pi}\int_{-\infty}^{\infty} \tilde{G}(k)\exp(ik\cdot x)dk.
\end{aligned}
\tag{20}
$$

Recognizing that $P(r_N)$ is a series of nested convolutions, we get:

$$
\begin{aligned}
P(r_N) &= \int_{-\infty}^{\infty} dr_1...\int_{-\infty}^{\infty} dr_{N-1}P(r_1,...,r_N) \\
A^{-1}P(r_N) &= \int_{-\infty}^{\infty} dr_1...\int_{-\infty}^{\infty} dr_{N-1}g(r_1)g(r_2-r_1)...g(r_N-r_{N-1}) \\
&= \int_{-\infty}^{\infty} dr_{N-1}...\int_{-\infty}^{\infty}\left[\int_{-\infty}^{\infty} dr_2\left[\int_{-\infty}^{\infty} dr_1 g(r_1)g(r_2-r_1)\right]g(r_3-r_2)\right]...g(r_N-r_{N-1}) \\
&= \mathcal{F}^{-1}\mathcal{F}\left[...\mathcal{F}^{-1}\left[\mathcal{F}\left[\mathcal{F}^{-1}[\mathcal{F}(g(r_1))\cdot\mathcal{F}(g(r_2-r_1))]\right]\mathcal{F}(g(r_3-r_2))\right]...\right]\mathcal{F}(g(r_N-r_{N-1})) \\
&= \mathcal{F}^{-1}[\mathcal{F}(g(r_1))\mathcal{F}(g(r_2-r_1))...\mathcal{F}(g(r_N-r_{N-1}))] \\
&= \mathcal{F}^{-1}[\tilde{G}(k)^N].
\end{aligned}
\tag{21}
$$

In the case of the Gaussian $g$ defined above :

$$
\begin{aligned}
\tilde{G}(k) &= \sqrt{2\pi b^2}\exp\left(-\frac{k^2b^2}{2}\right) \\
\tilde{G}(k)^N &= (2\pi b^2)^{N/2}\exp\left(-\frac{Nk^2b^2}{2}\right) \\
A^{-1}P(r_N) &= \mathcal{F}^{-1}[\tilde{G}(k)^N] \\
&= \frac{1}{2\pi}\int_{-\infty}^{\infty} dk\,(2\pi b^2)^{N/2}\exp\left(-\frac{Nk^2b^2}{2}\right)\exp(ik\cdot r_N) \\
&= (2\pi b^2)^{N/2}\sqrt{\frac{1}{2\pi Nb^2}}\exp\left(\frac{-r_N^2}{2Nb^2}\right),
\end{aligned}
\tag{22}
$$

so,

$$
P(r_N) = \sqrt{\frac{1}{2\pi Nb^2}}\exp\left(\frac{-r_N^2}{2Nb^2}\right).
\tag{23}
$$

Setting $N = s$, where $s$ is the chain contour length in numbers of monomers, the final contact probability of a linear Gaussian chain in 1D is:

$$
P_c(s) = P(r_N = 0) = (2\pi b^2 s)^{-\frac{1}{2}},
\tag{24}
$$

and in 3D it is:

$$
P_c(s) = (2\pi b^2 s)^{-\frac{3}{2}}.
\tag{25}
$$

This recovers standard results in polymer physics, and the classical $-3/2$ scaling coefficient for Gaussian polymer chains.

## Contact probability within a loop (circular chain)

In the case of contacts within a circular chain (i.e., a loop; *Figure 4—figure supplement 8i*), the chain configuration probability is built similarly, but is conditioned on the fact that the last chain segment must return to the first segment:

$$P(r_1, ..., r_N) = B\left[\prod_{i=1}^{N} g(r_i - r_{i-1})\right] g(r_N - r_0). \tag{26}$$

Again, this equation can be solved for the normalization factor $B$ using the Convolution Theorem and Fourier transforming procedure as above.

The distance probability distribution for the $s^{\text{th}}$ segment is given by:

$$P(r_s) = \prod_{i=1;i\neq s}^{N} \int_{-\infty}^{\infty} dr_i P(r_1, ..., r_N). \tag{28}$$

These integrals can also be solved by recognizing that we can use the Convolution Theorem separately by splitting the equation into two parts:

$$
\begin{aligned}
B^{-1}P(r_s) &= \left[\prod_{i=1}^{s} \int_{-\infty}^{\infty} dr_i g(r_i - r_{i-1})\right]\left[\prod_{i=s+1}^{N} \int_{-\infty}^{\infty} dr_i g(r_i - r_{i-1})g(r_N - r_0)\right] \\
&= \mathcal{F}^{-1}[\mathcal{F}(g(r_1))...\mathcal{F}(g(r_s - r_{s-1}))]\mathcal{F}^{-1}[\mathcal{F}(g(r_N - r_0))...\mathcal{F}(g(r_{s+1} - r_s))] \\
&= \mathcal{F}^{-1}\left[\tilde{G}(k)^s\right]\mathcal{F}^{-1}\left[\tilde{G}(k)^{N-s}\right] \\
&= (2\pi b^2)^{s/2}\sqrt{\frac{1}{2\pi s b^2}}\exp\left(\frac{-r_s^2}{2sb^2}\right) \cdot (2\pi b^2)^{(N-s)/2}\sqrt{\frac{1}{2\pi(N-s)b^2}}\exp\left(\frac{-r_s^2}{2(N-s)b^2}\right) \\
&= (2\pi b^2)^{N/2}\frac{1}{2\pi b^2\sqrt{s(N-s)}}\exp\left(-\frac{Nr_s^2}{2b^2 s(N-s)}\right)
\end{aligned}
\tag{29}
$$

So, we get for $P(r_s)$:

$$P(r_s) = \frac{\sqrt{N+1}}{2\pi b^2\sqrt{s(N-s)}}\exp\left(-\frac{Nr_s^2}{2b^2 s(N-s)}\right). \tag{30}$$

Thus, the contact probability of the $s^{\text{th}}$ segment (in 1D) is:

$$P_c = P(r_s = 0) = \frac{1}{2\pi b^2}\frac{\sqrt{N+1}}{\sqrt{s(N-s)}}. \tag{31}$$

In 3D, the solution is:

$$P_c = P(r_s = 0)^3 = \left(\frac{\sqrt{N+1}}{2\pi b^2\sqrt{s(N-s)}}\right)^3 \approx \left(\frac{1}{2\pi b^2\sqrt{s(1-s/N)}}\right)^3 = \left(\frac{1}{2\pi b^2\sqrt{s_{\text{eff}}}}\right)^3. \tag{32}$$

Interestingly, the genomic distance $s$ is replaced by the harmonic mean of the two paths within the loop. We can thus define an effective genomic distance $s_{\text{eff}}$ as $s_{\text{eff}} = s(1 - s/N)$.

## Contact probability between a loop and a linear segment

For a loop (circular chain) of total length $N$, connected to a linear chain segment of total length $L$ (*Figure 4—figure supplement 8ii*), the spatial distribution (in 1D) is given by:

$$P(r, s, L, N) = C\int_{-\infty}^{\infty} dr_s P_{\text{linear}}(r - r_s, L)P_{\text{circular}}(r_s - r_0, s, N). \tag{33}$$

The solution to this equation is:

$$P(r,s,L,N) = C(2\pi)^{\frac{L}{2}-1}b^{L-2}\sqrt{\frac{N+1}{NL+s(N-s)}}e^{-\frac{Nr^2}{2b^2(NL+s(N-s))}}, \tag{34}$$

where

$$C = (2\pi)^{\frac{1}{2}-\frac{L}{2}}\sqrt{\frac{N}{N+1}}b^{1-L}. \tag{35}$$

Then, the spatial distribution is:

$$P(r,s,L,N) = \frac{1}{\sqrt{2\pi b^2}}\sqrt{\frac{1}{L+s(1-\frac{s}{N})}}e^{-\frac{r^2}{2b^2(L+s(1-s/N))}}, \tag{36}$$

and the contact probability as a function of $s$, $N$, $L$ (in 1D) is thus:

$$P_c(r,s,L,N) = \frac{1}{\sqrt{2\pi b^2}}\sqrt{\frac{1}{L+s(1-\frac{s}{N})}} = \frac{1}{\sqrt{2\pi b^2}}\sqrt{\frac{1}{s_{\text{eff}}}}. \tag{37}$$

Here, the effective genomic distance $s_{\text{eff}} = L+s(1-s/N)$.

## Contact probability between chain segments with intervening loops

The contact probability of a chain with intervening loops (*i.e.*, loops that do not enclose the two points of interest) is simply calculated by ignoring the intervening loop. For instance, in a linear chain segment with one intervening loop of length $N$ (**Figure 4—figure supplement 8 iii**), the effective contact probability is $s_{\text{eff}} = s - N$.

## Contact probability between two connected loops

For the contact probability between any two connected loops (as in **Figure 4—figure supplement 8iv**):

$$P(r,s_1,N_1,s_2,N_2) = E\int_{-\infty}^{\infty}dr_{s_1}P_{\text{circular}}(r_{s_1}-r_0,s_1,N_1)P_{\text{circular}}(r-r_{s_1},s_2,N_2). \tag{38}$$

Similarly to the previous sections, this calculation yields:

$$P_c(s_1,N_1,r_l,L,s_2,N_2) = \frac{(2\pi b^2)^{-1/2}}{\sqrt{s_1(1-s_1/N_1)+s_2(1-s_2/N_2)}} = \frac{(2\pi b^2)^{-1/2}}{\sqrt{s_{\text{eff}}}} \tag{39}$$

In this case, the effective genomic distance is $s_{\text{eff}} = s_1(1-s_1/N_1)+s_2(1-s_2/N_2)$.

## Comparing semi-analytically generated contact maps to polymer molecular dynamics

We can readily generalize the above results to any configuration of loops on a polymer chain provided that the loops do not form pseudoknots. The 3D contact probability can be calculated between any two points of the polymer chain by:

$$P_c(s_{\text{eff}}) = \left(\frac{1}{2\pi b^2}\right)^3\left(\frac{1}{s_{\text{eff}}}\right)^{3/2}, \tag{40}$$

where $s_{\text{eff}}$ is obtained using the rules derived above. In summary, $s_{\text{eff}}$ is the effective shortest path between two points on the chain (computed by the sum of linear segments plus the harmonic means of 'looped'/circular chain segments). The above rules can be used to calculate the 'exact' looped Gaussian chain contact maps for any individual configuration of SMC

complex positions on the polymer chain. However, we can better approximate a Hi-C map (which is an average over a population of cells, each with a different configuration of SMC complexes) by subsampling from the full distribution of SMC configurations. An example of a map generated from such a subsampling method (which we refer to as the semi-analytical method) is shown in *Figure 4—figure supplement 6*, and it is compared to the contact map generated by an equivalent 3D polymer MD simulation.

These maps were generated for a circular chromosome of length 4000 monomers (where one monomer = 1 kb), with a single SMC complex loading site near the *ori* (position 0 kb). A total of 10 SMC complexes were randomly loaded on the chromosome, and they performed loop extrusion as outlined in the Materials and methods section in the main text. Contact maps were generated semi-analytically by using the SMC complex positions directly, or computed by real 3D contacts in an MD simulation with a cutoff contact-radius of 6 monomer lengths. As seen in *Figure 4—figure supplement 6*, the two calculated maps are visually very similar.

The differences between the semi-analytical and MD-simulated maps occur primarily at short genomic distances (<30 kb), where excluded volume interactions and the 3D polymer 'contact radius' play a role. However, for most of the genome, the semi-analytical and MD-simulation methods yield almost indistinguishable results for a short, bacterial chromosome as evidenced by the genome-wide contact probability curve (*Figure 4—figure supplement 7*).

