## [Decision Letter]

Thank you for submitting your article “Chromosome organization by one-sided and two-sided loop extrusion” for consideration by *eLife*. Your article has been reviewed by one peer reviewer, and the evaluation has been overseen by a Senior/Reviewing Editor. The reviewer has opted to remain anonymous.

The reviewers have discussed the reviews with one another and the Reviewing Editor has drafted this decision to help you prepare a revised submission. The required revision should address the issues raised in the review.

Reviewer #1:

This paper aims to resolve an important issue in the SMC protein field by reconciling conflicting results from in vitro single-molecule experiments that showed 1-sided DNA loop extrusion, while previous loop-extrusion based modelling of in vivo Hi-C data assumed 2-sided extrusion. From the extensive simulations presented in this paper, it is concluded that pure 1-sided motors cannot account for the full range of biological phenomena associated with loop extrusion, but various alternatives are offered that account for both the experimental observation of such motors and the in vivo data. This is interesting and new.

The manuscript is well structured and clearly articulated, and examines biologically relevant cases. The modelling from the Marko/Mirny labs has been instrumental for our understanding of DNA loop extrusion, and this paper again adds valuable new information. The conclusions are well supported by the simulation data and discussed in the context of biology and recent single molecule experiments.

I recommend publication of the work in *eLife* after the authors address the questions and comments mentioned below.

– In some of the modelling, a fair comparison with the data is obtained with the semi-diffusive 1D LE model. However this is realized for very large values of *v*_diff_, which can as large as (or exceeding) the LE speed v. With v≈ 1 kb/s, it seems entirely unreasonable to assume a *v*_diff_ of that same order of magnitude, as this would imply an extremely fast diffusion of the large SMC complexes along DNA. The authors should discuss how (un)reasonable large such values for *v*_diff_ are, and comment on this, if these are indeed physically implausible.

– The authors generally appear to conflate 'eukaryotic chromosomes' with 'mammalian chromosomes' (e.g. with statements such as 'Unlooped chromatin from one-sided extrusion hinders eukaryotic chromosome compaction and organization', etc). This seems unwarranted and should be distinguished more precisely. A major motivation for the current modelling work seems to be the in vitro work by Ganji et al., who measured 2D LE for yeast condensin from *S. cerevisiae*. Hence it would be natural to compare the simulation results with chromosomal compaction and HiC results obtained for yeast condensin. Hi-C results on yeast SMC proteins have yielded quite different behavior from that of metazoan cells (likely because of the relatively small chromosomes), and notably yeast does not have CTCFs which are a major boundary element in vertebrate cells. Indeed, budding yeast condensin seems to organize more stripes pattern than dots in HiC data (Figure 5B of Schalbetter et al., 2017). In addition, yeast cohesin showed different TAD organization patterns (Schalbetter et al., 2017; Tanizawa et al., 2017). The authors do have a brief paragraph describing some yeast compaction data at the end of their paper, but I would advise to include a more explicit modelling of the chromosomal compaction and HiC for yeast throughput the paper, as well as a much more in-depth discussion regarding the associated comparison to the published yeast data.

---

## [Author Response]

[…] I recommend publication of the work in eLife after the authors address the questions and comments mentioned below.– In some of the modelling, a fair comparison with the data is obtained with the semi-diffusive 1D LE model. However this is realized for very large values of v_diff_, which can as large as (or exceeding) the LE speed v. With v≈ 1 kb/s, it seems entirely unreasonable to assume a v_diff_ of that same order of magnitude, as this would imply an extremely fast diffusion of the large SMC complexes along DNA. The authors should discuss how (un)reasonable large such values for v_diff_ are, and comment on this, if these are indeed physically implausible.

The reviewer’s comment highlights two points: 1) the magnitude of diffusion that corresponds to diffusive stepping speeds of *v*_diff_~1 kb/s and 2) the ability of semi-diffusive LEFs to compact and organize chromatin.

First, the diffusion coefficient is given by *D* = *a v*_diff_, where *a* is the length of a lattice site (*a* = 0.5 kb, 2 kb, and 1 kb for the mitotic, interphase, and bacterial models, respectively). In our modeling, we study the dependence of the diffusive stepping speed by varying the dimensionless ratio *v*_diff_/*v* over the range 0.005 < *v*_diff_/*v* < 3.5. If we assume that the active subunit translocates at *v*=1 kb/s as observed in vitro, these values of *v*_diff_/*v* correspond to 0.0025 kb^2^/s < *D* < 7 kb^2^/s (accounting for the different values of *a*). On naked DNA in vitro, SMC complexes have been observed to diffuse with diffusion coefficients of 0.001 μm^2^/s < *D* < 4 μm^2^/s (or 0.01 kb^2^/s < *D* < 35 kb^2^/s), depending on the SMC complex, salt conditions, and the presence or absence of ATP (Davidson et al.,2016, Kanke et al., 2016, Kim and Loparo, 2016, Stigler et al., 2016, Terakawa et al., 2017, Kim et al., 2019). The diffusion coefficients in our model are within this range. Furthermore, we have studied a range of *D* that includes the limiting behaviors for both fast and slow diffusion coefficients.

Second, over this entire range of diffusion coefficients, we find that the semi-diffusive model *cannot* generate the chromosome organization phenomena of interest. In mitotic chromosomes, linear compaction is <100-fold, volumetric compaction is ≤2-fold, and sister chromatids do not spatially resolve (median separation is less than twice the backbone length; Figure 2D). In interphase, TAD bodies but not dots/corner peaks are observed (primary dot strength < 2; Figure 3D and Figure 3—figure supplement 2B). In bacteria, the arms of the circular chromosomes cannot be juxtaposed (Figure 4 and Figure 4—figure supplement 1).

We have made three sets of revisions to address the reviewer’s comment. First, we now provide the diffusion coefficients, *D*, measured in vitroand studied in the model in the “Model” section (“Semi-diffusive model” subsection), in the semi-diffusive subsection of the mitotic chromosomes section, and in the legends of Figures 3 and 4 (corresponding to the interphase and bacteria models). Second, we have revised the text in several places to clearly state that the semi-diffusive model fails for all studied scenarios. In the semi-diffusive subsection for the interphase model, we now refer to Figure 3—figure supplement 2, which quantifies dot strength. In the bacteria semi-diffusive subsection, we have added a clause clarifying that the semi-diffusive model fails for all values of *v*_diff_/*v*. In the second paragraph of the Discussion subsection “Unlooped chromatin from one-sided extrusion…”, we have added a similar clause. Third, we now remark in the second paragraph of the Discussion subsection “Unlooped chromatin from one-sided extrusion…” that the “safety-belt mutant” in Ganji et al., 2018, which is a possible experimental realization of semi-diffusive condensin, does not effectively grow loops, which is consistent with our conclusions. We believe that these revisions clarify that we have explored plausible diffusion coefficients and, moreover, that the semi-diffusive loop extrusion model fails for all chromosome organization scenarios that we considered.

– The authors generally appear to conflate 'eukaryotic chromosomes' with 'mammalian chromosomes' (e.g. with statements such as 'Unlooped chromatin from one-sided extrusion hinders eukaryotic chromosome compaction and organization', etc). This seems unwarranted and should be distinguished more precisely.

We thank the reviewer for pointing out this imprecise terminology. We have revised the manuscript where appropriate. We now generally use “metazoan” when describing the relevant scenario for mitotic chromosomes and “vertebrate” when describing the relevant scenario for interphase chromosomes. In a few places, we have kept the word “eukaryotic” because it is appropriately used.

A major motivation for the current modelling work seems to be the in vitro work by Ganji et al., who measured 2D LE for yeast condensin from *S. cerevisiae.* Hence it would be natural to compare the simulation results with chromosomal compaction and HiC results obtained for yeast condensin. Hi-C results on yeast SMC proteins have yielded quite different behavior from that of metazoan cells (likely because of the relatively small chromosomes), and notably yeast does not have CTCFs which are a major boundary element in vertebrate cells. Indeed, budding yeast condensin seems to organize more stripes pattern than dots in HiC data (Figure 5B of Schalbetter et al., 2017). In addition, yeast cohesin showed different TAD organization patterns (Schalbetter et al., 2017; Tanizawa et al., 2017). The authors do have a brief paragraph describing some yeast compaction data at the end of their paper, but I would advise to include a more explicit modelling of the chromosomal compaction and HiC for yeast throughput the paper, as well as a much more in-depth discussion regarding the associated comparison to the published yeast data.

We thank the reviewer for the excellent idea to consider our findings in the context of yeast chromosome organization. We chose the three scenarios of mitotic chromosome compaction, interphase TAD formation, and bacterial chromosomal arm juxtaposition because they encompass the main functions we associate with SMCs: linear and 3D compaction and spatial segregation, local domains formed by cis loops and linear scanning, and progressive juxtaposition of chromatin/DNA flanking a loading site. The revised manuscript explains this reasoning in the Introduction (third paragraph) and Discussion (second paragraph).

As a result, several major features of yeast chromosome organization were already indirectly considered by our analysis. A full set of simulations to match Hi-C observations in yeast (such as in Schalbetter et al. and Tanizawa et al., among others) would not provide additional insight beyond what has already been stated in our manuscript, Schalbetter et al., and Tanizawa et al. This is because loop extrusion in yeast is responsible for linear compaction and formation of chromatin domains, similar to what we have already studied for vertebrate cells. (Moreover, we note that *Xenopus* condensins also perform one-sided extrusion on naked DNA in mitotic cell extract (Golfier et al., 2019)). As such, information for specific systems (such as yeast) can be inferred from the existing analysis. Therefore, we have added a new subsection to the Discussion section, entitled “One-sided extrusion may be viable for yeast chromosomes in some, but not all, scenarios.” We briefly summarize its contents below.

In budding yeast (*S. cerevisiae*), cohesin compacts the genome during mitosis. From FISH and other microscopy (e.g., Guacci et al., 1994, Kruitwagen et al., 2018), we estimate that yeast is compacted ~10-fold. In fact, the modeling by Schalbetter et al. shows that Hi-C maps of mitotic chromosomes can be produced by cohesin-extruded loops that cover ~30-40% of the genome (*i.e.*, 2-fold compaction). One-sided extrusion can achieve this level of compaction (see Figure 2C(i) and newly added Figure 2—figure supplement 7).

Budding yeastcondensin, along with fission yeast (*S. pombe*) condensins and cohesins, generate features that are best understood through our analysis of local chromatin domains formed by loop extrusion. The features we consider are: 1) compaction of pre- and post-rDNA regions into domains by *S. cerevisiae* condensin in exponentially growing cells (Lazar-Stefanita et al., 2017, Schalbetter et al., 2017), 2) formation of small (<100 kb) domains by *S. pombe* cohesin (Mizuguchi et al., 2014, Kim et al., 2016, Tanizawa et al., 2017), 3) formation of large (100 kb – 1 Mb) domains by *S. pombe* condensin during mitosis (Kim et al., 2016, Kakui et al., 2017, Tanizawa et al., 2017), 4) formation of small (10-60 kb) condensin-dependent domains with dots (*i.e.*, bringing together boundaries) in quiescent *S. cerevisiae* (Swygert et al., 2019), and 5) formation of small domains with dots by overexpressed cohesin in growing *S. cerevisiae* (Dauban et al., 2020) or wild-type cells in S phase (Ohno et al., 2019).

Pure one-sided extrusion is compatible with features 1-3 because domains can be formed without dots (Figure 3B). Features 4 and 5 cannot be produced by pure one-sided extrusion with uniform loading alone because one-sided extrusion cannot reliably produce “dots” by bringing boundaries together (Figure 3B, C and Figure 3—figure supplement 2). Feature 4 suggests that *S. cerevisiae* condensin should be either effectively two-sided or one-sided but assisted by targeted loading or two-sided loop-extruding cohesins. Feature 5 suggests the same for *S. cerevisiae* cohesin, along with one additional possibility. Since dots formed via overexpressed cohesins are observed at sites convergent transcription, it is possible that one-sided extrusion in a “moving barrier” model (Brandão et al., 2019), in which RNA polymerase can push SMC complexes. By pushing the passive cohesin subunit, RNA polymerase could assist in generating dots at sites of convergent transcription. However, this mechanism requires that active genes around dots are oriented in an optimal way and that RNA polymerases efficiently push passive SMC complex subunits. We explain this argument in the main text and in the new Figure 3—figure supplement 8.